# The dyskerin ribonucleoprotein complex as an OCT4/SOX2 coactivator in embryonic stem cells

Yick W Fong[1†], Jaclyn J Ho[1], Carla Inouye[1], Robert Tjian[1,2]*

[1]Department of Molecular and Cell Biology, Howard Hughes Medical Institute, University of California, Berkeley, Berkeley, United States; [2]Li Ka Shing Center for Biomedical and Health Sciences, University of California, Berkeley, Berkeley, United States

**Abstract** Acquisition of pluripotency is driven largely at the transcriptional level by activators OCT4, SOX2, and NANOG that must in turn cooperate with diverse coactivators to execute stem cell-specific gene expression programs. Using a biochemically defined in vitro transcription system that mediates OCT4/SOX2 and coactivator-dependent transcription of the *Nanog* gene, we report the purification and identification of the dyskerin (DKC1) ribonucleoprotein complex as an OCT4/SOX2 coactivator whose activity appears to be modulated by a subset of associated small nucleolar RNAs (snoRNAs). The DKC1 complex occupies enhancers and regulates the expression of key pluripotency genes critical for self-renewal in embryonic stem (ES) cells. Depletion of DKC1 in fibroblasts significantly decreased the efficiency of induced pluripotent stem (iPS) cell generation. This study thus reveals an unanticipated transcriptional role of the DKC1 complex in stem cell maintenance and somatic cell reprogramming.

*For correspondence: tjianr@hhmi.org

Present address: †Brigham Regenerative Medicine Center, Cardiovascular Division, Department of Medicine, Brigham and Women's Hospital, Harvard Medical School, Cambridge, United States

## Introduction

The acquisition of pluripotency in the epiblast, a transient population of cells with unrestricted developmental potential during early embryogenesis, is controlled by a core set of transcription factors that include OCT4, SOX2 and NANOG (*Nichols et al., 1998*; *Avilion et al., 2003*; *Chambers et al., 2003*; *Mitsui et al., 2003*; *Silva et al., 2009*). This undifferentiated, pristine stem state can be captured as embryonic stem (ES) cells (*Evans and Kaufman, 1981*; *Martin, 1981*; *Brook and Gardner, 1997*), regenerated from somatic cells by cell fusion and nuclear transfer (*Yamanaka and Blau, 2010*), or by the ectopic expression of defined transcription factors (*Takahashi and Yamanaka, 2006*; *Yu et al., 2007*). These reprogrammed pluripotent cells display a transcriptome that is highly similar to ES cells. Not surprisingly, OCT4, SOX2 and NANOG also play key roles in the maintenance of pluripotency in ES cells and its reacquisition in induced pluripotent stem (iPS) cells by targeting a common set of genes that underpin the pluripotent state (*Boyer et al., 2005*; *Loh et al., 2006*; *Chen et al., 2008*). However, execution of these complex stem cell-specific gene expression programs also require a growing list of co-regulators including enhancer binding transcription factors (KLF4 (*Jiang et al., 2008*), SALL4 (*Wu et al., 2006*; *Zhang et al., 2006*), ESRRB (*Zhang et al., 2008*; *Festuccia et al., 2012*)), coactivators (Mediator (*Chia et al., 2010*; *Kagey et al., 2010*), YAP (*Lian et al., 2010*), TAFs/TFIID (*Fong et al., 2011*; *Liu et al., 2011*; *Pijnappel et al., 2013*)), chromatin remodelers (esBAF (*Ho et al., 2009*)), and histone modifiers (p300/CBP (*Chen et al., 2008*), the trithorax histone methyltransferase (*Ang et al., 2011*)). Perhaps the involvement of this rather elaborate collection of cofactors arose from the need for ES cells to significantly expand their transcriptional repertoire in order to accommodate the wide range of gene expression responses governing self-renewal and the transition

**eLife digest** The stem cells found in an embryo are able to develop into any of the cell types found in the body of the animal: an ability called pluripotency. When a cell becomes a specialized cell type, such as a nerve cell or a muscle cell, it loses this ability. However, mature cells can be reprogrammed back to a pluripotent state by artificially introducing certain proteins (known as 'reprogramming factors') into the mature cells.

A core group of reprogramming factors are known to activate networks of genes that are normally only expressed in stem cells, and by doing so trigger and maintain a pluripotent state. Other proteins help these core factors to regulate these networks of genes. In 2011, researchers discovered that a protein complex called XPC—which is normally involved in DNA repair—also helps two core reprogramming factors to activate an important gene related to pluripotency.

Now, Fong et al., including several of the researchers involved in the 2011 work, have identified another unexpected partner for the same two core reprogramming factors. The protein complex, called DKC1, has a number of known functions related to the processing of RNA molecules. This complex has also been linked to a fatal, rare human disorder called dyskeratosis congenita—a condition that affects many parts of the body, including the skin and bone marrow. Fong et al. found that when embryonic stems cells from mice are depleted of the DKC1 complex, the activation of important pluripotency-related genes by two of the core reprogramming factors is markedly reduced.

The XPC and DKC1 protein complexes were found to interact in pluripotent cells, and together they can activate a pluripotency-related gene to a greater extent than either can individually. Fong et al. propose that DKC1 binds to XPC, which in turn binds to two of the core reprogramming factors.

The DKC1 complex also binds to RNA molecules, and Fong et al. found that when the DKC1 complex binds to certain RNAs it is more able to help reprogramming factors activate pluripotency-related genes. On the other hand, other RNA molecules seem to inhibit the complex's ability to activate these genes.

Mutations identified in people with dyskeratosis congenita can prevent the DKC1 complex from binding to a subset of human RNA molecules. Moreover, the activity of stem cells is impaired in people with this developmental condition. As such, one of the next challenges will be to investigate if these mutations and RNA binding could be linked to problems with the activation of genes related to pluripotency in stem cells.

into diverse differentiated cell-types (*Fong et al., 2012*). Intriguingly, recent studies have implicated additional cofactors that have not been traditionally associated with transcriptional regulation, such as the XPC DNA repair complex (*Fong et al., 2011*), as well as microRNAs and long non-coding RNAs as part of the pluripotency regulatory network (*Wilusz et al., 2009*; *Orkin and Hochedlinger, 2011*; *Jia et al., 2013*).

Unconventional transcriptional coactivators like the XPC complex and YAP are often found to be multifunctional. For example, the XPC complex safeguards genome integrity of self-renewing stem cells as well as their differentiated progeny by scanning the genome for DNA damage and initiating excision repair (*de Laat et al., 1999*; *Riedl et al., 2003*; *Sugasawa, 2011*), while YAP controls the expansion of stem cells by sensing diffusible signals and external cues in the niche (*Lian et al., 2010*; *Dupont et al., 2011*; *Schlegelmilch et al., 2011*; *Mori et al., 2014*). It therefore seems reasonable to speculate that co-opting these protein complexes into performing gene regulatory functions may represent a prevalent evolutionary strategy that allows rapidly dividing stem cells to expand and enhance the pluripotency network while coping with the enormous pressure to maintain genome stability and cellular homeostasis. Indeed, coactivators like the XPC complex and YAP are highly enriched in ES and iPS cells perhaps because they are performing double duty (*Ramalho-Santos et al., 2002*; *Lian et al., 2010*; *Fong et al., 2011*). Not surprisingly, depletion of these multifaceted complexes compromises pluripotency gene expression, stem cell maintenance, and somatic cell reprogramming (*Lian et al., 2010*; *Fong et al., 2011*). Therefore, it appears that a critical threshold level of these coactivators may be required for a stem cell to maintain its pluripotency. Likewise, high levels of these

cofactors may be necessary to establish an appropriate gene regulatory environment for a somatic cell to re-enter the cell cycle and become responsive to transcription factor-mediated reprogramming.

Somatic cell reprogramming by a small cadre of specific transcription factors is thought to be a stochastic and inefficient process where only a small fraction (0.1–1%) of somatic cells become iPS cells (*Buganim et al., 2013*). However, recent data suggests that the induction of these rare, reprogramming-permissive somatic cells is not entirely a random event but may depend in part on cell-intrinsic determinants that are somehow restricted to a privileged few (*Guo et al., 2014*). These privileged somatic cells exhibit ultrafast cell duplication and express higher levels of proteins involved in DNA repair, RNA processing and cell cycle control (*Guo et al., 2014*). It is thought that these enrichments are required to fuel the rapid cellular proliferation necessary to overcome some major bottlenecks in reprogramming (*Banito et al., 2009*; *Hanna et al., 2009*; *Hong et al., 2009*; *Utikal et al., 2009*; *Ruiz et al., 2011*). Another roadblock to cellular reprogramming is the requisite early reactivation of a robust transcriptional circuitry governed by OCT4 and SOX2 (*Buganim et al., 2012*). Although this process can be enhanced by a number of transcription factors, reprogramming efficiency remains stubbornly low. It seems likely, therefore, that some key components of reprogramming remain undiscovered and there is a need to better define the molecular mechanisms by which OCT4 and SOX2 activate a stem cell-specific transcriptional program in ES and iPS cells.

To directly screen in an unbiased manner for cofactor requirements that support OCT4 and SOX2 mediated activation, we developed an in vitro transcription assay that faithfully recapitulates OCT4/SOX2 and coactivator-dependent gene activation observed in ES cells using purified components to reconstitute the human transcriptional apparatus (*Fong et al., 2011*). Deploying this sensitive biochemical complementation assay, we recently detected two additional stem cell coactivators (SCC-A and -B) that, in concert with the XPC coactivator complex, co-dependently stimulate the transcriptional activation of the *Nanog* gene by OCT4 and SOX2. Here we report that SCC-A activity is delivered by a subset of the dyskerin ribonucleoprotein complexes (DKC1 RNPs). We examined the specific activity of the various endogenous DKC1 RNPs assembled with distinct small nucleolar RNAs (snoR-NAs) by in vitro transcription. Furthermore, we combined promoter occupancy data with pluripotency gene expression profiles from loss-of-function studies to directly link the DKC1 complex to transcriptional coactivator function in ES cells. In addition to its well-documented role in regulating the proliferative capacity of stem cells, our studies unveil a previously unrecognized direct role of non-coding snoRNAs and the DKC1 complex in regulating transcription initiation with important implications for understanding the cell-intrinsic determinants conducive to cellular reprogramming.

## Results

### Purification and identification of Q0.3

We previously have shown an activity present in a partially purified protein fraction, Q0.3, that is required for the XPC coactivator complex to stimulate a full, synergistic activation of the human *Nanog* proximal promoter by OCT4 and SOX2 but is dispensable for basal or Sp1-activated transcription (*Rodda et al., 2005*; *Fong et al., 2011*). Q0.3 separated from the XPC complex at the Poros-HQ anion exchange chromatographic step (*Figure 1A,B*). Although Q0.3 appeared to migrate as a single activity on a size exclusion column with an apparent molecular mass ($M_r$) of ~500 kDa (*Figure 1C*), this coactivator activity splits again into two distinct chromatographic fractions on a Poros-Heparin (Poros-HE) cation exchanger. One cofactor (SCC-B) eluted at ~0.4 M KCl whereas the second activity (SCC-A) eluted at ~0.6 M KCl (*Figure 1D*). Taken together, it appears that at least three distinct stem cell coactivators (one being the XPC complex) are required to generate a full, OCT4/SOX2-dependent transcriptional response. Starting with nuclear extracts prepared from 400 L of a pluripotent embryonal carcinoma (EC) cell line NTERA-2 (NT2), we used the reconstituted transcription system supplemented with recombinant XPC complex, purified OCT4, SOX2, and a modified human *Nanog* template, to purify SCC-A over six successive chromatographic columns resulting in >30,000-fold increase in specific activity (*Figure 1A*). Silver staining of the peak Poros-HE purified fractions revealed a distinct pattern of four major polypeptides that consistently co-purified with SCC-A activity (*Figure 1E*). For the remainder of this report, we focused on the identification and functional characterization of SCC-A in vitro and in vivo.

To identify the polypeptides comprising the SCC-A complex, peak Poros-HE fractions were pooled, concentrated and separated by SDS-PAGE. Tryptic digestion of the four excised gel bands followed

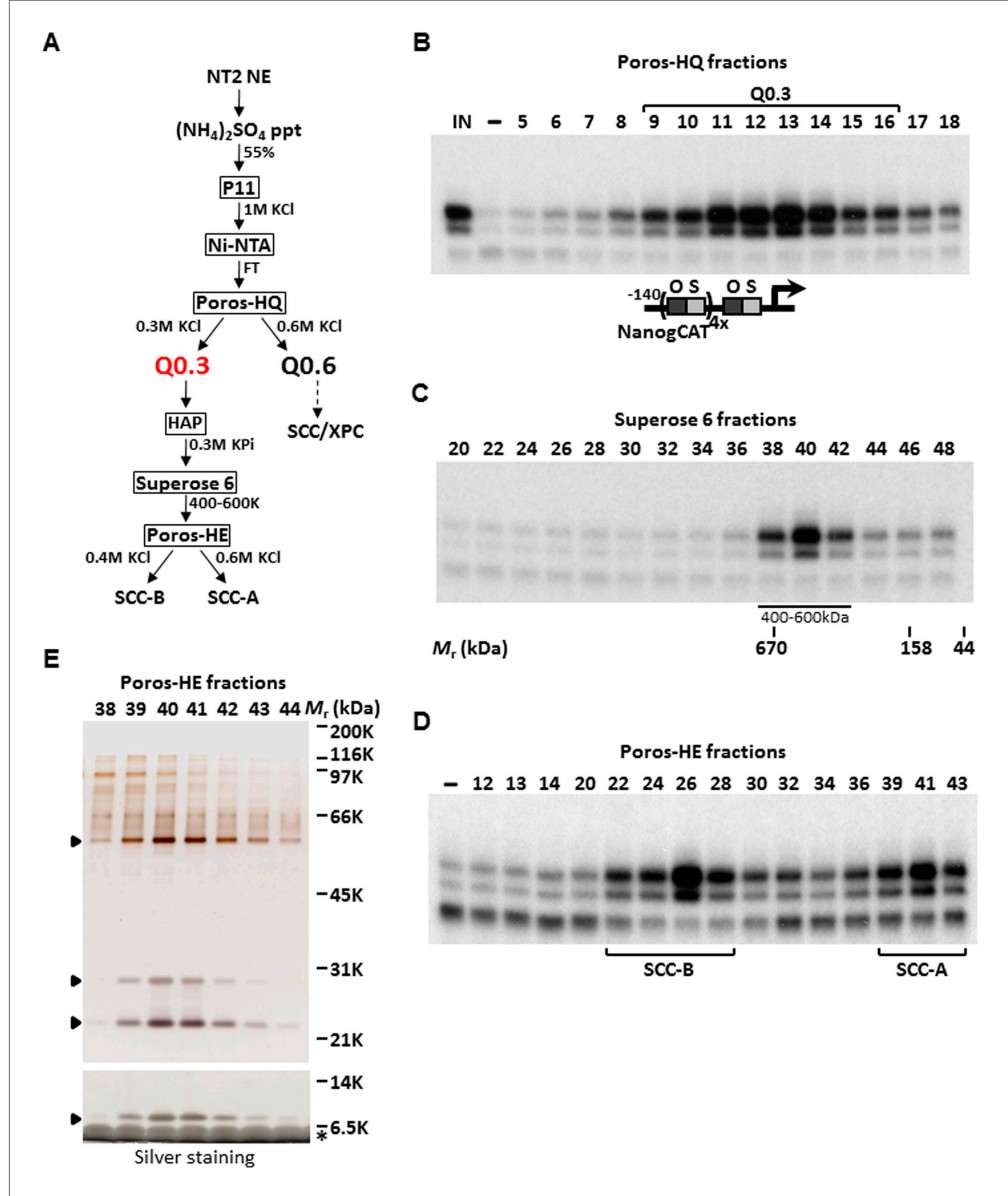

**Figure 1**. Purification of Stem Cell Coactivator-A (SCC-A) required for OCT4/SOX2-dependent activation of the *Nanog* gene. (**A**) Chromatography scheme for purification of Q0.3 from NT2 nuclear extracts (NT2 NE). NT2 NE is first subjected to ammonium sulfate precipitation (55% saturation) followed by a series of chromatographic columns including cation exchangers phosphocellulose (P11), heparin (Poros-HE), the anion exchanger Poros-HQ, hydroxyapatite (HAP), and gel filtration medium Superose 6. (**B**) Input (IN, Ni-NTA flowthrough), buffer control (−) and fractions containing Q0.3 eluted from a Poros-HQ anion exchanger (fraction number indicated) are assayed in the presence of OCT4, SOX2, and recombinant XPC complex in in vitro transcription assays. (**C**) Q0.3 appears to migrate as a single activity. Superose 6 fractions are assayed as in (**B**). Mobilities of peak activity (400–600 K) and gel filtration protein standards are shown at bottom. (**D**) Q0.3 is composed of two distinct coactivator activities, SCC-A and SCC-B. Transcription reactions contain buffer control (−), Poros-HE fractions and are assayed as in (**B**). SCC-A activity elutes in fractions 39–43. (**E**) Silver-stained 10% Bis-Tris polyacrylamide gel of the active SCC-A fractions. Filled arrowheads indicate polypeptides that co-migrate with SCC-A activity. The bottom panel shows the same fractions separated on a 12% SDS-PAGE gel to show the smallest subunit of SCC-A. Insulin added to Poros-HE fractions as a protein stabilizer is indicated by asterisk.

The following figure supplement is available for figure 1:

**Figure supplement 1**. The DKC1 and the XPC coactivator complexes are highly enriched in the transcriptionally active phosphocellulose 1 M KCl (P1M) and Ni-NTA flowthrough (Ni-FT) fractions.

by mass spectrometry analysis revealed SCC-A to be the dyskerin (DKC1) complex comprised of DKC1, GAR1, NHP2, and NOP10 subunits (*Figure 2A*) (*Meier, 2005*). Identification of the DKC1 complex as the active constituent of SCC-A activity was unexpected because it has not been previously linked to transcription. To corroborate the mass spectrometry data, we carried out western blot analysis to track the early chromatographic behavior of the three stem cell coactivators on a phosphocellulose column (*Figure 1A*). Consistent with our previous observation that the bulk of OCT4/SOX2 coactivator activity resides in the 1 M KCl fraction (P1M) (*Fong et al., 2011*), core subunits of the DKC1 and XPC complexes (and SCC-B, data not shown) were highly enriched in P1M compared to total nuclear extract or the transcriptionally inactive 0.3 and 0.5 M fractions (*Figure 1—figure supplement 1*).

## Reconstitution and mechanism of coactivation by the dyskerin complex

The DKC1 complex is an evolutionarily conserved, four-subunit protein complex that interacts with a large heterogeneous class of small non-coding RNAs called H/ACA small nucleolar RNAs (snoRNAs) (*Meier, 2005*; *Terns and Terns, 2006*). The assembly of a DKC1 RNP in vivo follows an elaborate, multi-step process mediated by the protein chaperones SHQ1 and NAF1 (*Darzacq et al., 2006*; *Grozdanov et al., 2009*). The GAR1 subunit subsequently replaces NAF1 in the intermediate complex containing NAF1, DKC1, NHP2, and NOP10 to form the mature RNP only after snoRNAs are incorporated and properly processed (*Kiss et al., 2010*). These H/ACA snoRNAs guide sequence-specific pseudouridylation of ribosomal RNAs (rRNAs) and spliceosomal small nuclear RNAs (snRNAs) by the catalytic subunit DKC1 (*Liang and Li, 2011*). The DKC1 complex also plays a key role in the biogenesis of telomerase by binding and promoting the processing and intranuclear trafficking of telomerase RNA (TERC) (*Egan and Collins, 2012*). Given the intimate association of the DKC1 complex with numerous RNAs and the multiple factors required to assemble the RNP in vivo, it is remarkable that an RNA-free, ternary 'apo-complex' can be generated in vitro. Indeed, several crystal structures of the archeal and yeast partial and holo-complexes of DKC1 revealed direct protein–protein contacts among the four subunits independent of RNA (*Li and Ye, 2006*; *Li et al., 2011*).

To firmly establish that the DKC1 complex rather than some trace contaminants present in the purified SCC-A fractions was responsible for the coactivator activity detected in our in vitro transcription assays, we reconstituted the human DKC1 complex from recombinant gene products expressed in insect (Sf9) and bacterial cells. Using a combination of conventional chromatography and affinity purification procedures, we were able to efficiently purify the recombinant DKC1 complex from Sf9 cells to near homogeneity (*Figure 2B*). It is important to point out that a significant amount of Sf9 RNAs co-purified with the human DKC1 complex as determined by 5′ end radiolabeling of the purified RNA species (*Figure 2—figure supplement 1*). This suggests that the biogenesis pathway and machinery for DKC1 RNP assembly are at least partly conserved between human and Sf9 cells. To determine whether specific snoRNAs are required for DKC1 coactivator function, we next attempted to reconstitute the DKC1 complex in *Escherichia coli*. However, the lack of dedicated chaperones (i.e. SHQ1 and NAF1) and accessory factors for the assembly of a DKC1 complex in *E. coli* made this task rather challenging, resulting in low yields after purification. Nonetheless, the holo-DKC1 complex isolated from *E. coli* showed similar subunit stoichiometry compared to DKC1 complexes purified from NT2 cells (*Figure 2A,C*). More importantly, it did not appear to contain detectable amounts of any associated RNAs (*Figure 2—figure supplement 1B*). To examine the contribution of individual subunits of the DKC1 complex in supporting OCT4/SOX2-activated transcription, we attempted to express and purify each of them individually in *E. coli*. However, all four gene products were either insoluble or remained tightly associated with bacterial heat shock proteins, suggesting that the individual protein subunits were not properly folded (data not shown). This is consistent with recent structural and functional analyses of free GAR1 and NOP10 showing that they are largely unfolded proteins (*Hamma et al., 2005*; *Li et al., 2011*). To circumvent this problem, we reconstituted partial complexes representing the different assembly intermediates during the biogenesis of DKC1 complexes in vivo in order to address the minimal protein subunit requirement for coactivator function (*Figure 2C*).

These hetero-dimeric (DKC1-NOP10), -trimeric (DKC1-NHP2-NOP10), ternary (NAF1-DKC1-NHP2-NOP10), and holo-DKC1 complexes were tested for their ability to potentiate OCT4/SOX2-dependent transcriptional activation of *Nanog* in vitro. Remarkably, all partial and complete recombinant complexes whether produced in *E. coli* or Sf9 cells exhibited similar specific activities for coactivation, but were reproducibly less active than the purified native endogenous DKC1 complex from NT2 cells

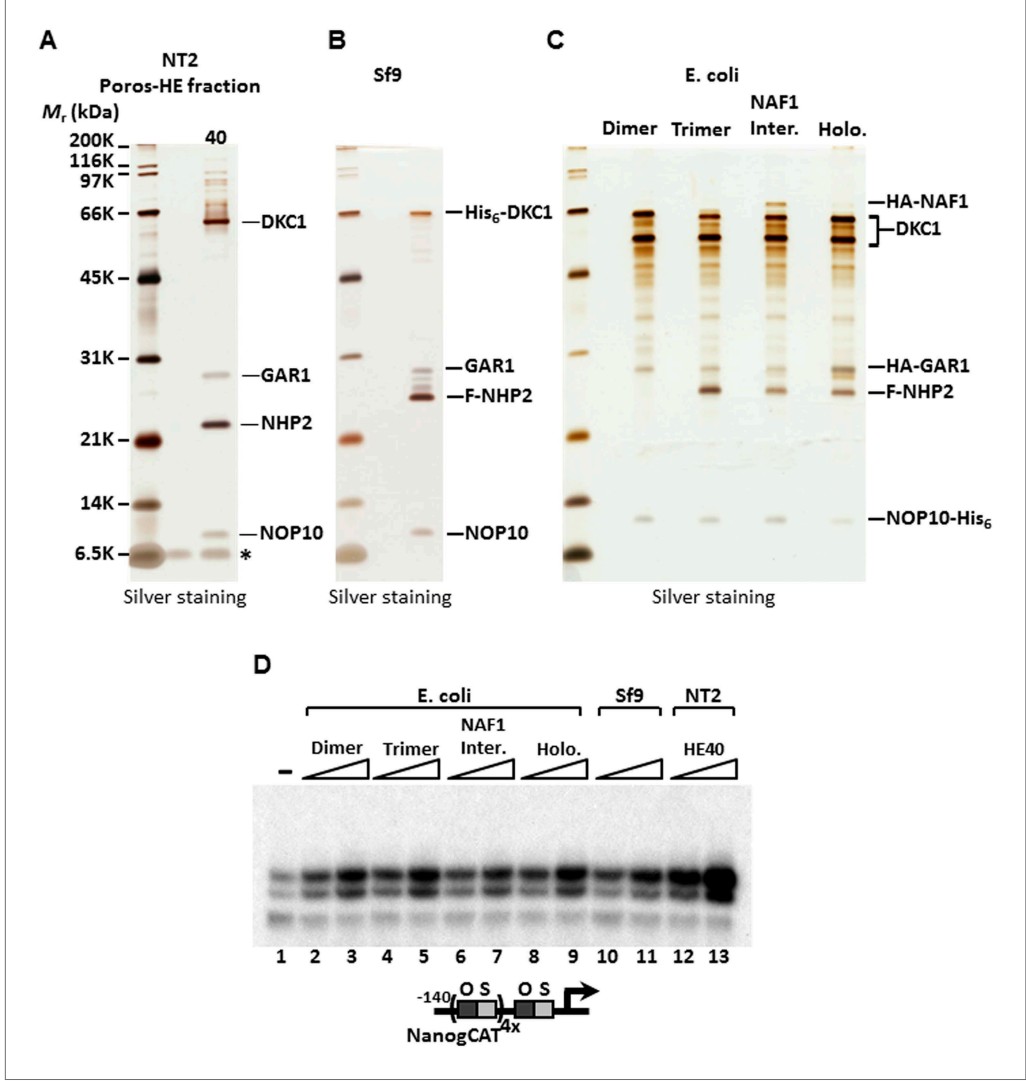

**Figure 2**. SCC-A is the dyskerin (DKC1) complex. (**A**) Silver stained SDS-PAGE gel of Poros-HE peak activity fraction in *Figure 1E* with protein identities determined by mass spectrometry analysis. (**B**) Silver stained SDS-PAGE gel of recombinant DKC1 complex reconstituted in insect Sf9 cells infected with baculoviruses expressing His₆-tagged DKC1, untagged GAR1, FLAG-tagged NHP2, and untagged NOP10. (**C**) Silver stained SDS-PAGE gel of recombinant partial and holo DKC1 as well as NAF1-containing intermediate complexes reconstituted in *E. coli* cells expressing various epitope-tagged subunits and untagged DKC1 as denoted. A prominent partial fragment of DKC1 co-purifies extensively with the full length DKC1 complexes. (**D**) Recombinant DKC1 complexes enhance OCT4/SOX2-activated transcription of *Nanog*. Buffer control (−), bacterial DKC1-NOP10 heterodimer (lanes 2 and 3), DKC1-NHP2-NOP10 trimer (lanes 4 and 5), NAF1-DKC1-NHP2-NOP10 intermediate (lanes 6 and 7), and holo DKC1-GAR1-NHP2-NOP10 (lanes 8 and 9), recombinant holo DKC1 complex purified from Sf9 cells (lanes 10 and 11), and endogenous holo-complex from NT2 (Poros-HE peak activity fraction 40, lanes 12 and 13) are assayed over a twofold concentration range. Transcription reactions contain OCT4, SOX2, recombinant XPC complex, and a Poros-HE fraction containing SCC-B.
The following figure supplement is available for figure 2:

**Figure supplement 1**. 5′ end radiolabeling of RNAs co-purified from recombinant DKC1 complexes.

---

(*Figure 2D*). It was not clear whether the reduced specific activities of the recombinant purified com-
plexes resulted from poorly folded or assembled subunits, presence of inhibitory RNAs, or both.
Nevertheless, these results using purified recombinant subunits confirm that at least the protein com-
ponents of the DKC1 complex represent a major contributor of the SCC-A coactivator function.
Indeed, it appears that the largest subunit DKC1 and the smallest protein NOP10 are sufficient to

provide the bulk of the transcriptional coactivator function and that an RNA component may not be strictly required for this moonlighting activity of the DKC1 complex. Although snoRNAs may not be essential for conferring coactivator competence to the recombinant DKC1 complexes, we note that the endogenous DKC1 complexes are twofold to threefold more active than their recombinant counterparts, suggesting that some mammalian snoRNAs may play a role in enhancing the transcriptional activity of the DKC1 complex.

## Some snoRNAs may modulate DKC1 coactivator function

The DKC1 RNPs in mammalian cells are highly heterogeneous—with more than 100 known H/ACA snoRNAs that form an equally large number of distinct RNPs by associating with the same four core protein subunits, some of which with unknown functions (i.e. orphan snoRNAs that lack base complementarity to rRNAs or snRNAs) (*Kiss et al., 2010*). New classes of snoRNAs have also been identified and shown to directly participate in disparate cellular processes from pre-mRNA splicing to chromatin decondensation (*Jady et al., 2012*; *Schubert et al., 2012*; *Yin et al., 2012*). Furthermore, as much as 60% of snoRNAs can be processed into microRNAs (miRNAs), most of which have unknown targets (*Ender et al., 2008*; *Taft et al., 2009*). Thus, our understanding of the full repertoire of H/ACA snoRNAs and their 'non-canonical' functions remains limited. It is also unclear if the binding of different snoRNAs to the human DKC1 complex induces structural changes or masks protein surfaces that may positively or negatively impact coactivator function. Given that most, if not all, GAR1-containing DKC1 complexes are mature RNPs in vivo (*Kiss et al., 2006*), it seemed prudent for us to examine the range and specific activity of these native but heterogeneous mixtures of human DKC1 RNPs.

Even though these 100 or more DKC1 RNPs have highly similar if not identical core protein composition and architecture, we reasoned that these RNPs are likely to display distinct chromatographic properties due to their unique snoRNAs and/or associated factors. In an attempt to biochemically fractionate this heterogeneous population of DKC1 RNPs, a partially purified fraction prepared from 200 L of NT2 cells that contains >95% of the total population of human DKC1 RNPs (Ni-FT; *Figure 1— figure supplement 1*) was applied to a Poros-HQ anion exchange column and fractionated using a salt gradient (*Figure 3A*). As expected, DKC1 RNPs were found to elute in a broad profile from 0.3 to 0.9 M KCl with the majority of the complexes eluting at ~ 0.5 M (*Figure 3B*), consistent with extensive heterogeneity of the DKC1 RNPs in NT2 cells. We next immuno-affinity purified the various DKC1 RNPs from different salt eluted Poros-HQ fractions using a monoclonal antibody against human DKC1 followed by peptide elution. The various affinity-purified DKC1 RNP pools all contain stoichiometric amounts of the four core protein subunits indicating that they are likely mature RNPs (*Figure 3C*). However, we failed to detect any other major associated polypeptides in these purified samples. Therefore, differences in protein composition alone are unlikely to fully account for the observed chromatographic heterogeneity of the DKC1 RNPs separated by the salt gradient on a Poros-HQ column. Instead, we strongly suspect the differential chromatographic behavior of the endogenous human DKC1 RNP complexes to derive from association with distinct RNA species. Indeed, 5' end radiolabeling of the purified RNA species from the various affinity-purified DKC1 RNP preparations revealed distinct patterns of associated RNAs (*Figure 3D*). The DKC1 RNPs purified from high salt eluted fractions (# 22, 26, and 30) were enriched for longer RNAs (>180 nucleotides) and some select shorter RNAs (<100 nucleotides). The 130–140 nucleotide-long snoRNA clusters were recovered from DKC1 immunoprecipitates from multiple fractions spanning a wide spectrum of the salt gradient. Thus, it appeared that parameters in addition to RNA length may contribute to the observed differential chromatographic properties of different DKC1 RNPs. Of note, the DKC1 RNPs purified from fraction 9 did not appear to contain significant amounts of RNA (*Figure 3D*). This is unexpected because the presence of GAR1 usually signifies that some RNA species should have been loaded into the complex in the normal course of DKC1 RNP assembly. However, we cannot exclude the possibilities that, although unlikely, RNAs were present but somehow refractory to labeling at both 5' (*Figure 3D*) and 3' ends (data not shown). It remains possible that some snoRNAs were degraded or had dissociated from a small fraction of the DKC1 RNPs during purification.

These highly purified pools of DKC1 RNPs were assayed over a threefold dose–response range in our fully reconstituted in vitro transcription reactions containing OCT4, SOX2, recombinant XPC complex and SCC-B. Remarkably, DKC1 RNPs purified from higher salt eluted Poros-HQ fractions (fractions 26 and 30) displayed significantly higher specific activity than those from lower salt fractions (fractions 9 and 14) (*Figure 3E*). We estimated a ~sixfold enhancement in the specific activities of

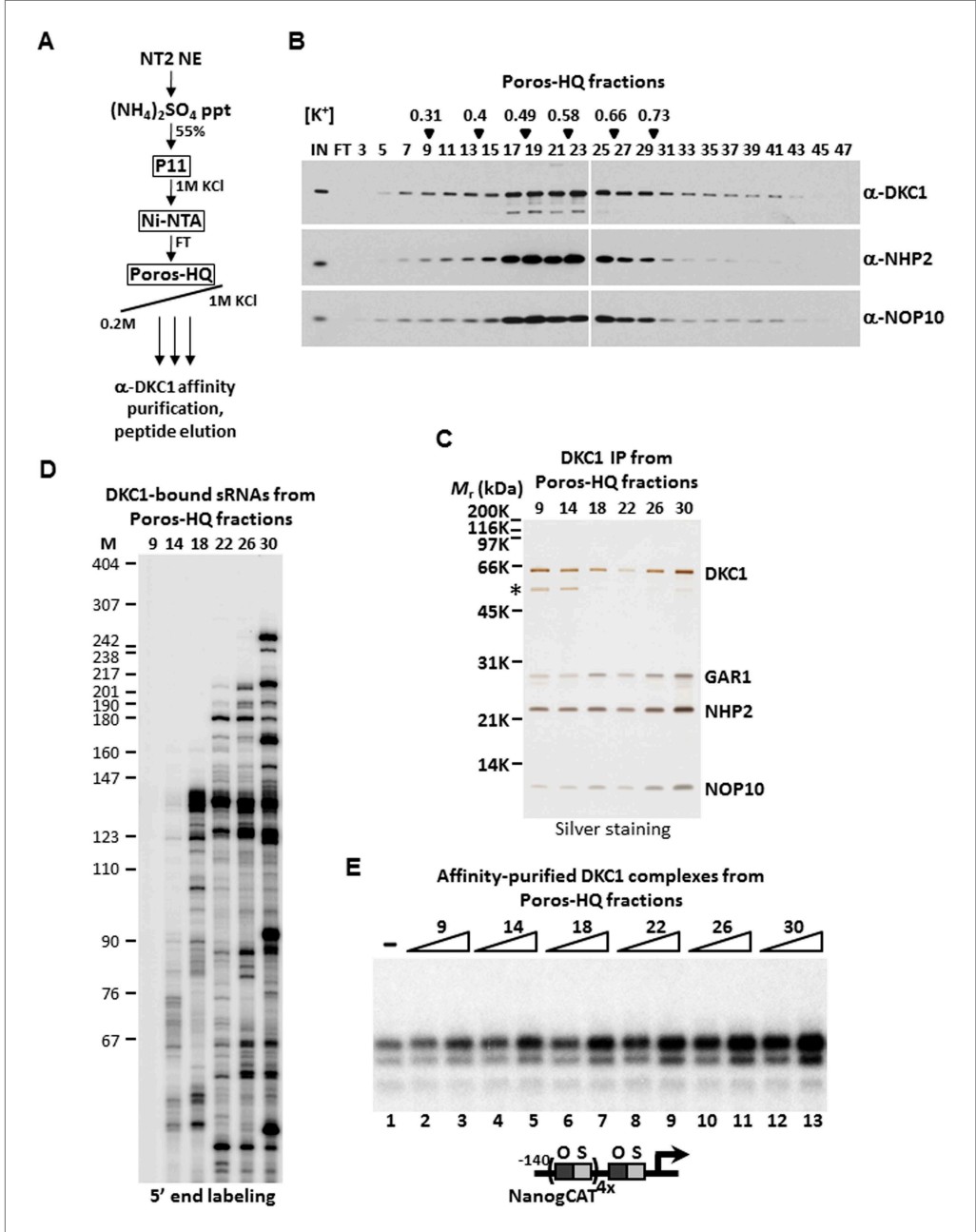

**Figure 3**. DKC1-associated small RNAs modulate transcriptional coactivator activity. (**A**) Purification scheme of endogenous DKC1 ribonucleoprotein complexes (RNPs). A partially purified fraction (Ni-FT) containing the bulk of the DKC1 RNPs in NT2 cells (See *Figure 1—figure supplement 1*) is fractionated over a Poros-HQ anion exchange column followed by affinity purification using a monoclonal antibody against DKC1 and peptide elution. (**B**) Extensive heterogeneity of the endogenous DKC1 RNPs from NT2 cells. Western blotting of input (IN), flow-through (FT), and various salt-eluted Poros-HQ fractions using antibodies against DKC1, NHP2, and NOP10. Filled inverted triangles indicate fraction numbers used for affinity purification. Salt concentrations ($[K^+]$ in M) of selected fractions are indicated. (**C**) Silver stained SDS-PAGE gel of the DKC1 RNPs affinity-purified from indicated Poros-HQ fractions. A proteolytic fragment of DKC1 is denoted by asterisk. (**D**) 5′ end labeling of RNAs isolated from affinity-purified DKC1 RNPs from indicated Poros-HQ fractions. Radiolabeled RNAs were separated on a 6% denaturing urea-polyacrylamide gel. Size markers are in nucleotides. (**E**) Buffer control (−) or affinity-purified DKC1 RNPs from salt-eluted Poros-HQ fractions over a threefold concentration range are assayed using in vitro transcription. Reactions contain OCT4, SOX2, recombinant XPC complex, a Poros-HE fraction containing SCC-B.

DKC1 RNPs purified from fraction 30 compared to fraction 9, which, as we had shown in *Figure 3D*, contained no detectable RNAs (*Figure 3E*, compare lanes 3 and 13). It is unclear if this endogenous apo-complex lacking any detectable RNA component is physiologically relevant or an experimental artifact. However, the fact that this apo-complex activated transcription with reduced specific activity (*Figure 3E*, compare lanes 1 and 3) is consistent with our previous observation that the bacterial apo-complex is less active than DKC1 RNPs purified from NT2 cells in supporting transcription (*Figure 2D*). Paradoxically, recombinant DKC1 RNPs purified from Sf9 cells contained insect snoRNAs (*Figure 2—figure supplement 1*) but exhibited low specific activities similar to the bacterial and apo-complexes suggesting that some RNAs may be inhibitory. Taken together, these results uncover a previously unrecognized gene regulatory role of the DKC1 RNP complex wherein a subset of mammalian non-coding snoRNAs may enhance the DKC1 coactivator function while other RNAs may inhibit its transcription activity.

## Mechanisms of coactivation in vitro and in vivo

Identification of the DKC1 RNP and the XPC DNA repair complexes as co-dependent coactivators for OCT4/SOX2 was unexpected on two fronts. These two multi-subunit protein assemblies had not been previously implicated in directing stem cell-specific transcription nor had they been functionally linked to each other in any cellular processes. We therefore set out to determine the functional relationship between these newly identified stem cell coactivators and their mechanisms of coactivation in vitro and in vivo. Our ability to recombinantly express and purify these coactivators (including purified SCC-B which will be the subject of a future study) allowed us to systematically test the contribution of each coactivator alone in supporting OCT4/SOX2-activated transcription in vitro. Addition of individual coactivator complexes only marginally activated *Nanog* transcription (*Figure 4A*). However, when the DKC1 complex was supplemented with the XPC complex, we observed a noticeable increase in transcriptional output that was substantially further enhanced by adding the third coactivator, SCC-B (*Figure 4A*). These results confirmed the co-dependent nature of these three coactivators in supporting an optimal, synergistic activation of the *Nanog* gene by OCT4 and SOX2. To further explore the mechanism by which the DKC1 complex cooperates with the XPC complex in OCT4/SOX2 activated transcription, we co-expressed both complexes along with (or without) OCT4 and SOX2 in 293T cells and performed co-immunoprecipitation assays to probe for a potential interaction between these two coactivators. Immunoprecipitation of the XPC complex pulled down the DKC1 complex both in the presence and absence of the activators. This finding suggests that the DKC1 complex may function as an OCT4/SOX2 coactivator in part through a direct physical interaction with the XPC complex, which in turn binds OCT4 and SOX2 (*Figure 4B*). In support of this observation, a recent global proteomic study using large scale biochemical fractionation of human cell extracts to isolate stable protein complexes identified WDR79, a known accessory protein of the mature DKC1 RNP (*Tycowski et al., 2009*; *Jady et al., 2012*), as a candidate XPC-interacting protein (*Havugimana et al., 2012*). Whether the DKC1 complex also forms direct contacts with OCT4 and SOX2 in the absence of XPC is unclear. Our attempt to address this was hampered by the fact that we could not express any of the four subunits of the DKC1 complex to a significant level in 293T or several other cell lines (data not shown). However, the fact that co-expression of OCT4/SOX2 did not increase the amount of DKC1 pulled down by the XPC complex argues against a stable tripartite complex wherein the coactivators interact with each other and form independent contacts with the activators.

Mutations in the *Dkc1*, *Nhp2*, and *Nop10* genes have been linked to dyskeratosis congenita (DC), a rare but fatal human genetic disorder that impairs stem cell function and proliferation generally attributed to defects in telomerase or ribosome biogenesis (*Mitchell et al., 1999*; *Mason and Bessler, 2011*). Our discovery of a stem cell-specific transcriptional role of the DKC1 complex adds a potentially important alternative mechanism for interpreting the molecular basis of DC disease phenotypes. However, it was unclear if amino acid residues critical for telomerase and ribosome biogenesis impinge on distinct or overlapping domains with respect to our newly uncovered DKC1 transcription coactivator function. To begin to address this potentially important link to disease, we focused on DC mutations in the large DKC1 subunit and the small NOP10 protein because a partial complex of these two subunits was sufficient to activate *Nanog* transcription in vitro (*Figure 2D*). We recombinantly expressed and purified a panel of mutant DKC1 complexes in Sf9 cells that are representative of both position (L37del (*Heiss et al., 1998*), A353V (*Knight et al., 1999*), Δ22C (*Vulliamy et al., 1999*)) and frequency (A353V) at which DC mutations occur in *Dkc1* (*Marrone et al., 2005*). We also generated an

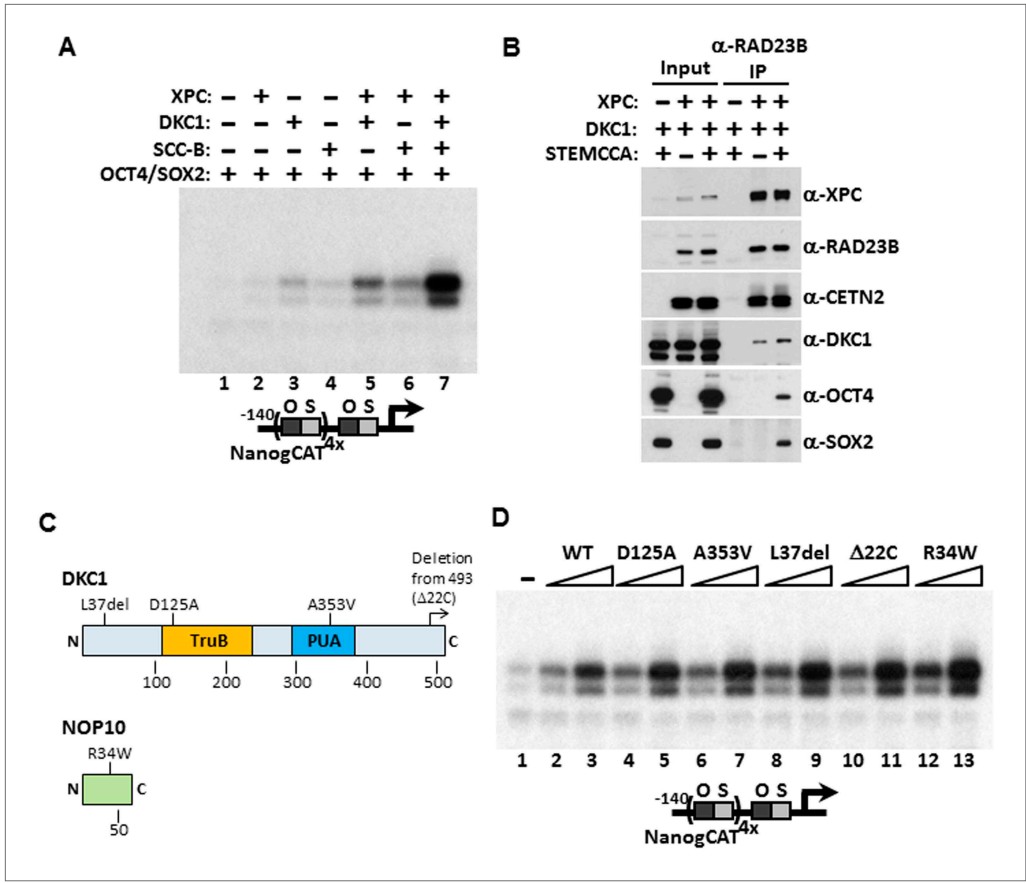

**Figure 4**. Mechanism of Coactivation by the DKC1 Complex. (**A**) Co-dependent activation of *Nanog* transcription by the XPC complex, the DKC1 complex, and SCC-B. Recombinant XPC and DKC1 complexes purified from Sf9 cells, and SCC-B purified from bacteria were added individually (lanes 2–4), or in various combinations (lanes 5–7) to in vitro transcription reactions containing OCT4 and SOX2. Strongest synergistic activation is observed when all three coactivators are present in the transcription reaction (lane 7). (**B**) DKC1 interacts with the XPC complex independent of OCT4 and SOX2 in vivo. Control (−), plasmids expressing mouse XPC complex (XPC), mouse DKC1 complex (DKC1), and STEMCCA were co-transfected into 293T cells. Cell lysates are immunoprecipitated with anti-RAD23B antibody. Input extracts (2%) and RAD23B-bound proteins were analyzed by western blotting. (**C**) Schematic diagrams showing the two structural domains in DKC1 (TruB and PUA) and mutations in DKC1 and NOP10 selected for functional analyses in (**D**). All mutations except D125A are identified in patients with dyskeratosis congenita (DC). (**D**) Wild type, pseudouridine synthase inactive (D125A), and DC mutant DKC1 complexes (*Dkc1* A353V, *Dkc1* L37del, *Dkc1* Δ22C, and *Nop10* R34W) are reconstituted in Sf9 cells and assayed over a threefold concentration range in in vitro transcription reactions containing OCT4, SOX2, recombinant XPC complex, and SCC-B.

The following figure supplements are available for figure 4:

**Figure supplement 1**. Micrococcal nuclease (MNase)-treated recombinant DKC1 complexes remain structurally intact.

**Figure supplement 2**. DKC1-associated RNAs in recombinant DKC1 complexes are resistant to extensive MNase digestion.

**Figure supplement 3**. MNase digestion moderately increases DKC1 coactivator activity.

artificial, pseudouridine synthase inactive mutant DKC1 (D125A (*Gu et al., 2013*)) as well as a mutant NOP10 containing (R34W (*Walne et al., 2007*)) complex (*Figure 4C*; *Figure 4—figure supplement 1*). Remarkably, all mutant DKC1 RNPs, whether they were mock or nuclease-treated to partially remove the associated Sf9 small RNAs (*Figure 4—figure supplement 2*), were consistently more active than

the WT holo-complex in potentiating OCT4/SOX2-mediated transcription (*Figure 4D*; *Figure 4—figure supplement 3*). Therefore, it appeared that neither the enzymatic activity nor amino acids mutated in DC are essential for coactivator activity although the enhanced coactivator phenotype could lead to changes in gene expression and altered stem cell function. The transcriptional phenotypes of these DKC1 mutations are highly reminiscent of our findings with the XPC complex in that disease-relevant amino acids and domains critical for DNA repair functions were also dispensable for OCT4/SOX2-activated transcription (*Fong et al., 2011*).

To further probe the molecular mechanisms by which the DKC1 complex might function as a transcriptional coactivator for OCT4 and SOX2 in ES cells, we performed chromatin immunoprecipitation (ChIP) assays to investigate whether the DKC1 complex is directly recruited to regulatory regions of key OCT4/SOX2-target genes. We found that efficient crosslinking of DKC1 to the *Oct4* enhancer by formaldehyde requires the pre-treatment of ES cells with a protein–protein cross-linker (ethylene glycol bis[succinimidylsuccinate] or EGS) (*Figure 5—figure supplement 1*). ChIP-qPCR analysis revealed that sites of DKC1 occupancy at the *Oct4*, *Nanog*, *Sox2* genes indeed coincide with those of OCT4 (*Boyer et al., 2005*; *Loh et al., 2006*) and SOX2 binding to enhancer and promoter DNA sequences (*Figure 5—figure supplement 2*) in the mouse ES cell line D3 (*Figure 5A*). Importantly, DKC1 binding is also enriched at the enhancers of *Oct4* and *Nanog* in human ES cell line H9 (*Figure 5B*) and EC cell line NT2 (*Figure 5C*), thus confirming the generality of a co-recruitment mechanism to transcriptional regulatory elements in pluripotent stem cells. Curiously, we failed to detect a significant enrichment of DKC1 at some OCT4/SOX2-target genes such as *Fgf4* in D3 cells (*Figure 5A*). This suggests that the DKC1 complex may be differentially employed by OCT4 and SOX2 to regulate a subset of their target genes. Additional experiments such as genome-wide analyses of DKC1 occupancy will be required to ascertain the extent to which DKC1 associates with OCT4/SOX2 target genes in mouse ES cells. Since over 90% of snoRNAs are embedded in the introns of coding and non-coding genes (*Filipowicz and Pogacic, 2002*), the DKC1 complex has also been found to localize at gene bodies where it is thought to co-transcriptionally process nascent snoRNAs (*Darzacq et al., 2002*; *Ballarino et al., 2005*; *Yang et al., 2005*). Now our finding of the DKC1 complex co-occupying pluripotent gene promoters and enhancer elements with sequence-specific activators OCT4 and SOX2 in ES cells strongly suggests a classical coactivator function of the DKC1 complex rather than acting purely as a snoRNP maturation factor.

## The DKC1 function in stem cell maintenance and somatic cell reprogramming

Many transcriptional activators (OCT4, SOX2, NANOG) and coactivators (Mediator, TAFs/TFIID, the XPC complex) critical for stem cell pluripotency are often highly enriched in ES cells but become rapidly down-regulated upon differentiation. Dynamic regulation of these transcription factors in ES cells is thought to confer not only stability to the transcriptional circuitry governing self-renewal but also the flexibility to exit the pluripotent state and switch between competing developmental programs during differentiation (*Jaenisch and Young, 2008*; *Liu et al., 2011*; *Fong et al., 2012*). Consistent with the notion that the DKC1 complex is performing as a stem cell-specific coactivator in ES cells, the DKC1, GAR1, and NOP10 subunits are highly enriched in pluripotent D3 cells (*Figure 6A*). Their levels in ES cells decreased rapidly upon retinoic acid (RA)-induced differentiation while general transcription factor TFIIB and loading control β-Actin remained unchanged. Importantly, the selective decrease of DKC1 levels was not simply a reflection of a reduced proliferative state or protein translational activity in differentiating ES cells because components of the C/D snoRNP (FBL and NOP58), another major machinery involved in the ribosome biogenesis pathway, stayed largely constant (*Su et al., 2013*). Indeed, it has been shown that transcription of the *Dkc1* gene is regulated by OCT4 and NANOG in ES and iPS cells (*Agarwal et al., 2010*), thus providing a transcriptional mechanism whereby *Dkc1* expression levels are tightly coupled to the pluripotent state.

To gain additional in vivo evidence that the DKC1 complex is required for the proper expression of genes critical for stem cell self-renewal, we performed loss-of-function studies using lentiviruses expressing two independent short hairpin RNAs (shRNAs) specifically targeting DKC1 in mouse D3 ES cells (*Figure 6B*). We also depleted XPC in D3 cells using a previously characterized shRNA (*Fong et al., 2011*) to investigate potential functional interactions between these two coactivator complexes. Interestingly, knockdown of DKC1 but not XPC resulted in co-depletion of the small NOP10 subunit indicating that the stability of individual subunits likely depends on the integrity of the DKC1 complex

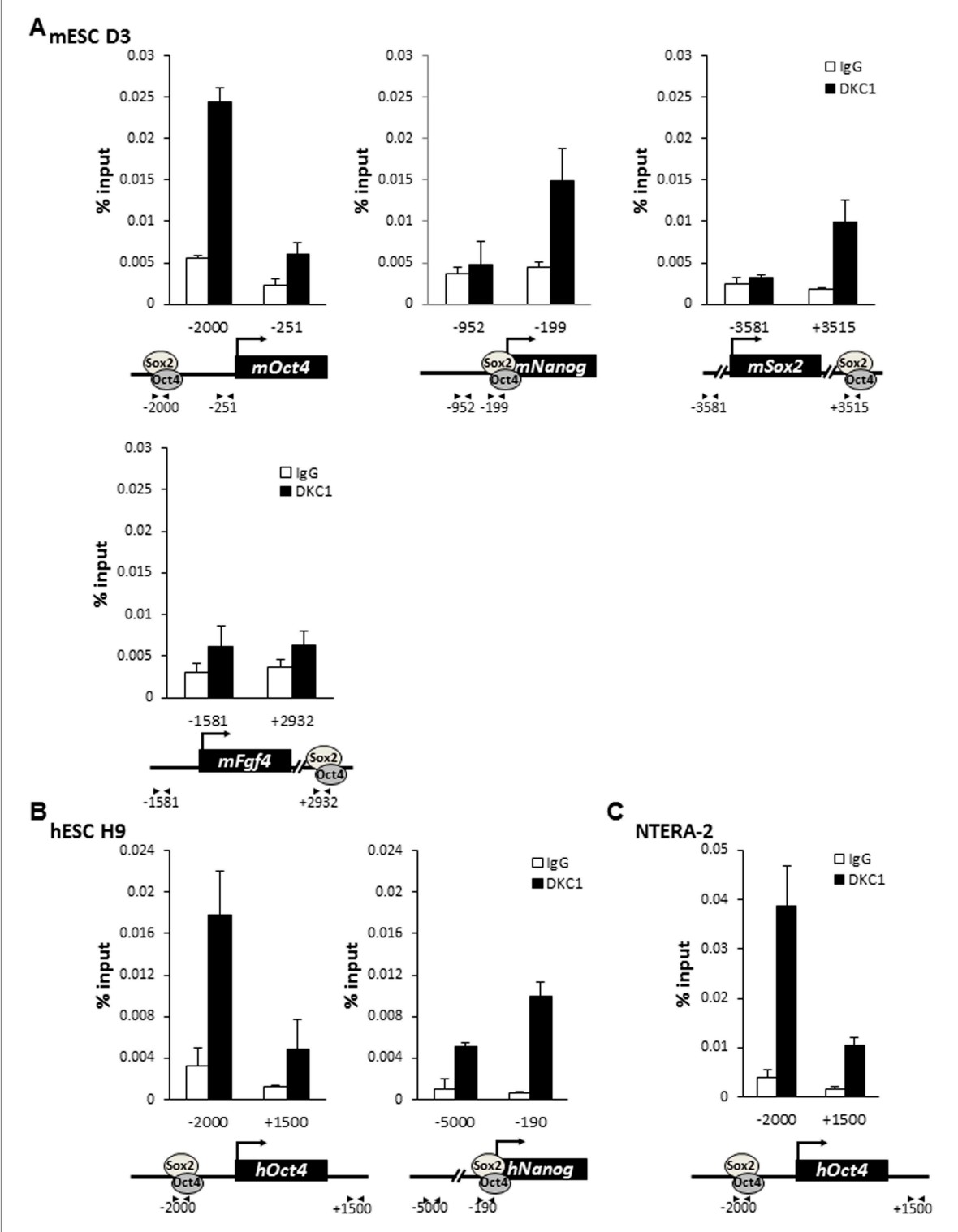

**Figure 5**. The DKC1 complex is recruited to regulatory regions of key pluripotency genes in mouse and human ES cells. (**A**) Co-occupancy of DKC1, OCT4, and SOX2 on enhancers of *Oct4*, *Nanog*, *Sox2*, but not *Fgf4*, in mouse ES cell line D3. Chromatin immunoprecipitation (ChIP) analysis of DKC1 occupancy on control and enhancer regions of the *Oct4*, *Nanog*, *Sox2*, and *Fgf4* gene loci. Representative data (n = 3) showing the enrichment of DKC1 (black bars) compared to control IgGs (white bars) are analyzed by qPCR and expressed as percentage of input chromatin. Schematic diagrams of OCT4/SOX2 binding sites of each gene and the relative positions of the amplicons used to detect enriched ChIP fragments are shown at the bottom.
*Figure 5. Continued on next page*

*Figure 5. Continued*

Error bars represent standard deviation, n = 3. Primer sequences can be found in *Supplementary file 1*. (**B**) DKC1 is recruited to regulatory regions of *Oct4* and *Nanog* in human ES cell line H9. Representative ChIP data (n = 3) are analyzed as described in (**A**). Error bars represent standard deviation, n = 3. (**C**) DKC1 is enriched on *Oct4* promoter in human embryonal carcinoma cell line NT2. Representative ChIP data (n = 3) are analyzed as described in (**A**). Error bars represent standard deviation, n = 3.

The following figure supplements are available for figure 5:

**Figure supplement 1**. Crosslinking DKC1 to chromatin requires protein–protein crosslinker ethylene glycol bis[succinimidylsuccinate] (EGS) in addition to formaldehyde (FA).

**Figure supplement 2**. SOX2 is enriched on the regulatory regions of *Oct4*, *Nanog*, and *Fgf4* in mouse ES cells.

---

(*Figure 6B*). This may also explain why a decrease in protein levels of GAR1 and NOP10 during RA-induced ES cell differentiation follows the same kinetics as DKC1 even though *Gar1* and *Nop10* do not appear to be direct targets of OCT4 and SOX2 (*Figure 6A*). Compared to control knockdown D3 cells, shRNA-mediated silencing of XPC (shXPC) or DKC1 (shDKC1-1 and shDKC1-2) resulted in pronounced morphological abnormalities including rapid collapse of the tightly packed ES cell colonies and appearance of large, flattened cells with concurrent dramatic reductions in alkaline phosphatase activity, all indicative of enhanced spontaneous differentiation of ES cells (*Figure 6C*). At this point, we cannot rule out the possibility that the severe phenotype observed in DKC1 knockdown ES cells is at least partially contributed by disruption of other well documented DKC1-dependent cellular processes (telomerase function and ribosome biogenesis) in addition to its transcription coactivator function. However, mouse ES cells lacking telomerase activity (*Terc* −/− (*Niida et al., 1998*)) or carrying a pathogenic mutation in *Dkc1* (A353V (*Mochizuki et al., 2004*)) can be maintained in culture for over 300 population doublings with no observable impact on growth rate and only a very mild effect on ribosome biogenesis. Since the self-renewal defects observed in DKC1 knockdown ES cells became apparent by 3 days post lentiviral infection (<9 population doublings), cellular senescence or a gross defect in rRNA processing are unlikely to be major contributors to the DKC1 knockdown phenotypes we observed in ES cells.

Consistent with the evident morphological changes associated with compromised stem cell identity, single knockdown of XPC or DKC1 in D3 cells resulted in a significant reduction in mRNA levels of core pluripotency genes including *Nanog*, *Oct4*, *Sox2*, *Klf4*, as well as stem cell marker *Fgf4* (*Figure 6D*), while housekeeping gene *Gapdh* remain stable (*Figure 6—figure supplement 1*). Interestingly, simultaneous knockdown of XPC and DKC1 did not further reduce their expression. This is consistent with the co-dependent nature of the DKC1 and XPC complexes in gene activation wherein the absence of one coactivator severely limited the ability of the other two stem cell coactivators to stimulate *Nanog* transcription in vitro (*Figure 4A*). To further explore the spontaneous differentiation phenotype in DKC1 and XPC-deficient ES cells, we performed qPCR analyses to monitor the expression level of lineage-specific markers representing the three germ layers and the trophectoderm. Depletion of DKC1 or XPC upregulated the expression of neuroectodermal maker *Fgf5* and trophoblast-specifier *Cdx2* at the expense of *Gata6*, a primitive endoderm marker, while mesodermal marker *T* remained unchanged (*Figure 6E*). Double knockdown of DKC1 and XPC appeared to further augment the expression of *Cdx2* but not *Fgf5*. The observed differentiation bias in DKC1 and XPC knockdown ES cells may be in part due to the reduced levels of OCT4 and NANOG, both of which have well-documented functions in antagonizing differentiation of extraembryonic lineages including the trophectoderm (*Niwa et al., 2000*; *Hay et al., 2004*; *Hyslop et al., 2005*; *Silva et al., 2009*).

The essential role of the DKC1 complex in establishing an OCT4/SOX2-dependent gene expression program in ES cells led us to hypothesize that DKC1 may be required for the reacquisition of pluripotency during cellular reprogramming by ectopic expression of OCT4, SOX2, KLF4, and c-MYC (*Takahashi and Yamanaka, 2006*). Of note, recent studies showed that primary human adult fibroblasts (HFs) carrying pathogenic mutations in *Dkc1* are refractory to cellular reprogramming (*Agarwal et al., 2010*; *Batista et al., 2011*). However, it is important to point out several key differences between using MEFs and HFs derived from DC patients to study DKC1 function in somatic cell reprogramming. Unlike MEFs which display high levels of telomerase activity and long telomeres (>50 kb (*Blasco et al., 1997*)),

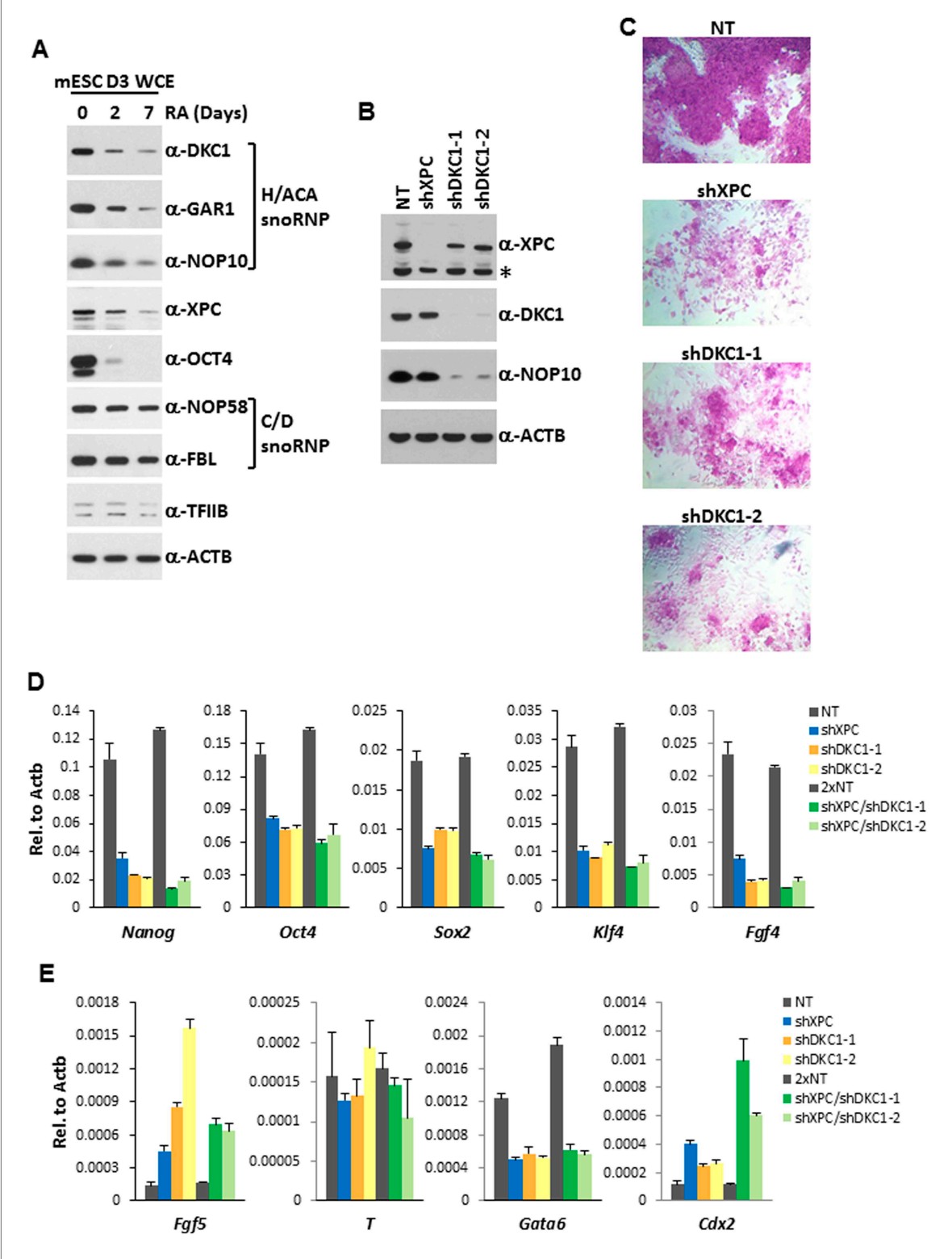

**Figure 6**. The DKC1 complex is required for stem cell maintenance. (**A**) Downregulation of the DKC1 complex upon retinoic acid (RA)-induced differentiation of mouse ES cell line D3. Western blot analyses of whole cell extracts prepared from D3 cells (mESC D3 WCE) collected at indicated days post LIF withdrawal and RA treatment using antibodies against the DKC1 complex (DKC1, GAR1, and NOP10), XPC, OCT4, the NOP58/fibrillarin (FBL) complex, TFIIB, and β-actin as loading control. (**B**) shRNA-mediated knockdown of the DKC1 complex in mouse ES cells. Whole cell extracts of mouse D3 cells

*Figure 6. Continued on next page*

*Figure 6. Continued*

infected with control non-target (NT) lentiviruses or with lentiviruses targeting XPC (shXPC) or DKC1 (shDKC1-1 and shDKC1-2) are analyzed by western blotting. MOI = 25. Asterisk denotes non-specific signals. (**C**) ES cell colony morphology and alkaline phosphatase (AP) activity are maintained in control non-target shRNA infected D3 cells (NT), but are compromised in XPC (shXPC) and DKC1 depleted cells using two independent shRNAs (shDKC1-1 and shDKC1-2). (**D**) DKC1 and/or XPC depletion in ES cells compromised pluripotency gene expression. Quantification of *Nanog, Oct4, Sox2, Klf4*, and *Fgf4* mRNA levels in single and double knockdown of XPC and DKC1 in D3 cells are analyzed by qPCR and normalized to β-actin (*Actb*). For double knockdown experiments, a cumulative MOI = 50 is used. Data from representative experiments are shown. Error bars represent standard deviation (n = 3). (**E**) DKC1 and/or XPC depletion in ES cells induces spontaneous differentiation towards primitive ectoderm and trophectoderm. Quantification of mRNA levels of primitive ectoderm marker *Fgf5*, mesoderm marker *T*, primitive endoderm marker *Gata6*, and extraembryonic trophectoderm marker *Cdx2* in single and double knockdown of XPC and DKC1 in D3 cells are analyzed as in (**D**). Primer sequences can be found in **Supplementary file 1**.
The following figure supplement is available for figure 6:

**Figure supplement 1**. shRNA-mediated knockdown of DKC1 in mouse ES cells does not compromise housekeeping gene expression.

HFs lack measurable TERT activity and have relatively short telomeres (10–15 kb (*Harley et al., 1990*)). In fact, telomerase null MEFs can be propagated in culture for more than 200 cell divisions without loss of viability (*Blasco et al., 1997*), which make MEFs a potentially better cell culture system for studying telomerase-independent functions of DKC1 in reprogramming. By contrast, DC patient-specific fibroblasts have shorter telomeres and could also accumulate secondary mutations due to genome instability, which are both detrimental to the reprogramming process (*Fong et al., 2013*). Consistent with this notion, it was shown that ectopic expression of wild type DKC1 (or TERT) in a DC mutant fibroblast line (*Dkc1* L37del) failed to rescue the reprogramming defect phenotype (*Agarwal et al., 2010*). Therefore, it remains unclear what impact, if any, acute DKC1 depletion in MEFs will have on iPS cell generation.

To address this question, we infected MEFs with lentiviruses expressing non-targeting control shRNA or two independent shRNAs specific for DKC1 and initiated reprogramming by doxycycline (dox)-induced expression of OCT4, KLF2, SOX2 and c-MYC (OKSM) (*Sommer et al., 2009*). We observed a marked decrease in the number of AP-positive iPS cell colonies (~20–50-fold reduction) whether or not we plated the induced DKC1 knockdown MEFs directly onto gelatin coated plates (where the surrounding DKC1 knockdown MEFs refractory to reprogramming acted as feeder cells) or onto mitomycin-treated feeder cells (*Figure 7A*; *Figure 7—figure supplement 1*). This suggests that failure of DKC1-deficient MEFs to acquire pluripotency is likely a cell autonomous phenomenon. Flow cytometry analysis showed that the majority of both control and DKC1 knockdown cells down-regulated fibroblast-associated cell surface marker THY1 indicating a loss of MEF identity (*Figure 7B*). However, unlike control cells where many of them became SSEA1+ and ultimately gave rise to AP and NANOG-positive iPS cell colonies, most DKC1 knockdown cells do not (*Figure 7B,C*). Because of the observed early arrest in reprogramming associated with DKC1-depleted MEFs, we next asked whether these cells were able to undergo the mesenchymal-to-epithelial transition (MET), a requisite initiating event prior to expression of SSEA1 antigen (*Li et al., 2010*; *Samavarchi-Tehrani et al., 2010*; *Golipour et al., 2012*; *Polo et al., 2012*). By day 14 post dox-induction, control knockdown MEFs showed reduced expression of fibroblast-enriched, pro-mesenchymal genes *Slug* and *Snail*, but their levels remained noticeably higher than those in ES cells (*Figure 7D*). This is likely due to contaminating partially reprogrammed iPS cells and residual fibroblasts present in the induced cell culture (*Figure 7B*). These non-target knockdown cells also acquired epithelial characteristics indicated by elevated levels of *Ecad* and *Epcam* (*Figure 7D*), as expected, given that THY1-/SSEA1+ partially and fully reprogrammed iPS cells represent the bulk of these control cells (*Figure 7B*). By contrast, depletion of DKC1 in MEFs blocked the reactivation of epithelial genes (*Ecad* and *Epcam*) without significantly perturbing the silencing of mesenchymal genes (*Figure 7D*), thus effectively uncoupling the otherwise tightly coordinated MET induced by OKSM (*Liu et al., 2013*). These data taken together suggest that DKC1 could be required for reprogrammed MEFs to acquire an epithelial identity during the critically important mesenchymal-to-epithelial transition.

To address whether the early reprogramming arrest observed in DKC1-depleted MEFs can be attributed to a gross defect in cellular proliferation, we labeled control and DKC1 knockdown MEFs with a stable dye (CFSE). The doubling time of these cells was measured by monitoring the decrease in dye intensity resulting from cell division over a 4 day period (*Figure 8A*). Although a lengthening of

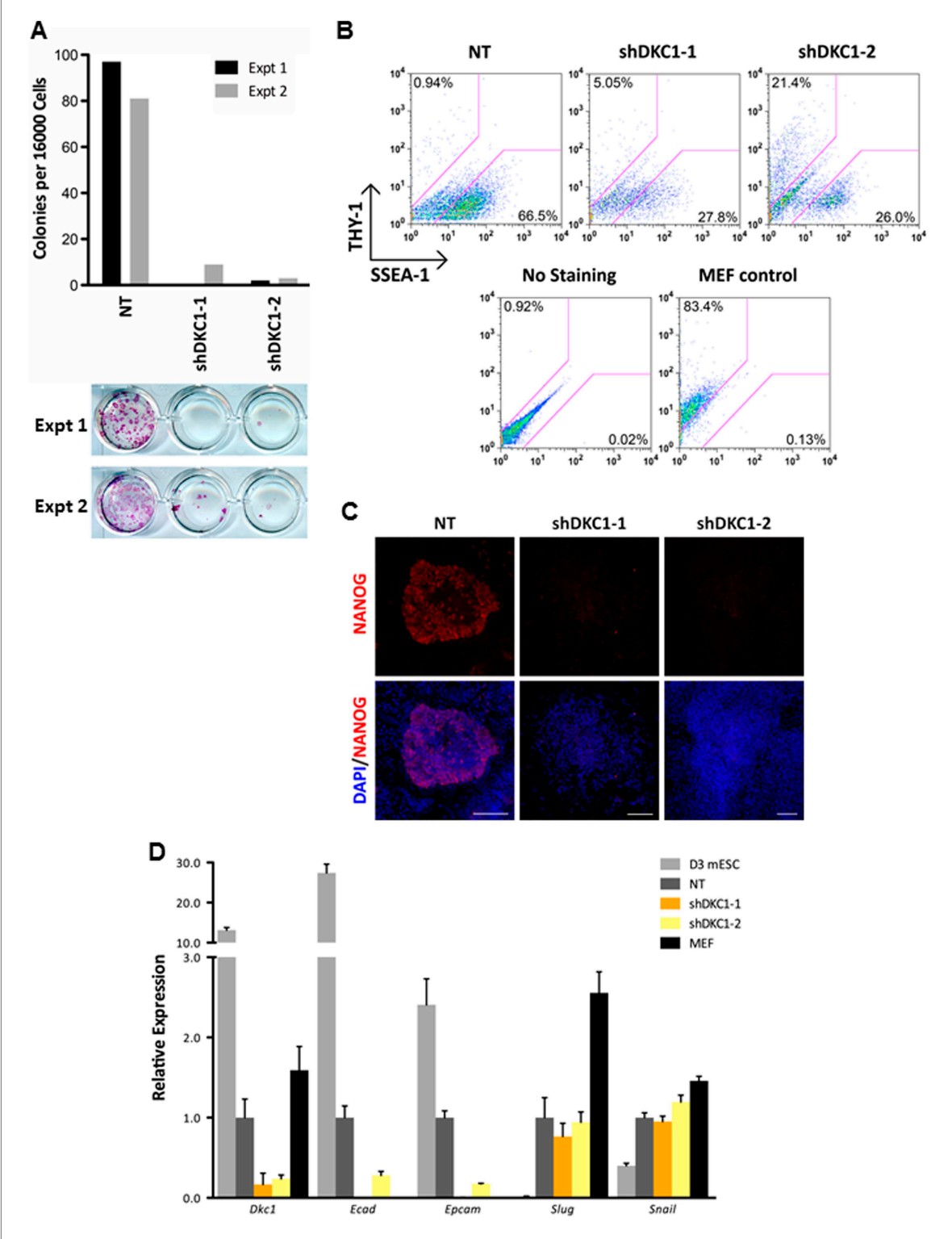

**Figure 7**. The DKC1 complex is required for mesenchymal-to-epithelial transition (MET) during somatic cell reprogramming. (**A**) Depletion of DKC1 blocks somatic cell reprogramming. CF-1 mouse embryonic fibroblasts (MEFs) are infected with lentiviruses expressing OCT4, KLF4, SOX2, and c-MYC (STEMCCA) and reverse tetracycline-controlled transactivator (rtTA) together with control non-target shRNA (NT) or two independent shRNAs targeting DKC1 (shDKC1-1 and shDKC1-2). Infected MEFs are plated onto gelatin coated 24-well plates (Experiment 1) or 24-well plates containing mitomycin-treated feeder MEFs (Experiment 2); cellular reprogramming is initiated by the addition of doxycycline (dox). Cells are stained for AP activity and

*Figure 7. Continued on next page*

*Figure 7. Continued*

counted after 14 days (11 days with dox followed by 3 days without dox) post induction (dpi). (**B**) Single cell suspensions of 14 dpi CF-1 MEFs as described in (**A**) are stained with anti-mouse SSEA-1 and anti-THY-1 antibodies and analyzed by flow cytometry. (**C**) Representative confocal images of NANOG stained colonies 17 dpi (14 days with dox followed by 3 days without dox) as described in (**A**). The same acquisition settings—excitation laser intensity, gain, and exposure time—were used for all NANOG images. Scale bar, 100 µm. (**D**) DKC1-depleted MEFs are arrested at the MET during iPS cell generation. Somatic cell reprogramming of CF-1 MEFs is performed as in (**A**). Cells were collected at 14 dpi (11 days with dox followed by 3 days without dox). mRNA levels of *Dkc1*, epithelial markers *Ecad* (also known as *Cdh1*) and *Epcam*, and mesenchymal markers *Slug* and *Snail* are compared with that of uninduced MEFs and mouse ES cell line D3 by qPCR. Values are normalized to expression levels in control non-target knockdown samples. Error bars represent standard deviation (n = 3). Primer sequences can be found in *Supplementary file 1*.

The following figure supplement is available for figure 7:

**Figure supplement 1**. Somatic cell reprogramming is blocked by DKC1 depletion.

doubling time of DKC1-depleted MEFs by both targeting shRNA hairpin was observed, compared to shDKC1-1, knockdown by shDKC1-2 has a significantly more pronounced effect on iPS cell generation while having a minimal impact on cellular proliferation (*Figure 7—figure supplement 1*; *Figure 8A*). This suggests that reprogramming efficiency does not strictly correlate with changes in proliferation rates caused by DKC1 depletion. However, we cannot exclude the possibility that differences in doubling rates could be due to differential off-target effects of the two hairpins. We also note that DKC1 depletion does not cause abrupt growth arrest of all MEFs but appears to selectively impair the proliferation of the faster cycling subpopulation without affecting the rest of the slower-dividing MEFs (*Figure 8A*). Therefore, factors in addition to growth impairment are at least contributing to the observed defect in somatic cell reprogramming. These observations are also consistent with recent findings suggesting that essentially all of the reprogramming potential in OKSM-induced MEF cultures is confined to a small fraction (1–8%) of cells characterized by an ultrafast cell cycle (*Smith et al., 2010*; *Guo et al., 2014*). Given the multiple functions of the DKC1 complex in regulating cellular proliferation (*Alawi et al., 2011*), MET (*Figure 7D*) and pluripotency gene expression (*Figure 6D*), we asked whether DKC1 might also be involved in overcoming barriers to deterministic cellular reprogramming. After 4 days of dox-induced expression of OKSM in MEFs, we labeled cells with CFSE and continued dox treatment for another 2 days before subjecting them to FACS (*Guo et al., 2014*). We identified and characterized four distinct cell populations bearing variegated dye concentrations (*Figure 8B*). The fastest dividing population (CFSE-Lo) had undergone at least 4 more cell divisions than the bulk MEFs and gave rise to substantially more AP-positive iPS cell colonies than the slower-dividing populations (*Figure 8B*; *Figure 8—figure supplement 1*). Strikingly, CFSE-Lo cells also expressed the highest levels of *Dkc1* reaching that of ES cells (*Figure 8C*). These cells have lost their mesenchymal identity and initiated the transition into cells of epithelial origin (*Figure 8D*). By contrast, the slower dividing populations expressed significantly lower levels of *Dkc1* and failed to fully silence mesenchymal genes or robustly reactivate epithelial markers indicating a delayed or abortive MET. Importantly, using MEFs carrying an integrated dox-inducible OKSM expression cassette (*Carey et al., 2010*), we observed a similar preferential enrichment of *Dkc1* in the fastest-dividing population (CFSE-Lo) despite uniform *Oct4* expression levels among CFSE-Lo, Med, and Hi cells (*Figure 8—figure supplement 2,3*). Therefore, an early onset of MET appears to be a defining property of these ultrafast cycling cells wherein appropriately high levels of DKC1 are necessary and likely serve as an important gene-specific transcriptional coactivator.

## Discussion

Our de novo identification of the DKC1 complex as a transcriptional coactivator for OCT4/SOX2 underscores the expanding repertoire of this multifunctional ribonucleoprotein complex (RNP) in stem cells. Beyond its well-documented role in ribosome and telomerase biogenesis, the DKC1 complex has been shown to effect diverse cellular processes including internal ribosome entry site (IRES)-dependent translation (*Yoon et al., 2006*) and base excision of 5-hydroxymethyluridines in rRNA by uracil-DNA glycosylase 1 (SMUG1) (*Jobert et al., 2013*). Interestingly, the telomerase complex itself has been implicated in the regulation of MYC and WNT/β-catenin associated gene expression programs critical for stem cell function (*Choi et al., 2008*; *Park et al., 2009*). However, the reverse transcriptase TERT, but curiously not its catalytic activity, was reported to be required for gene activation

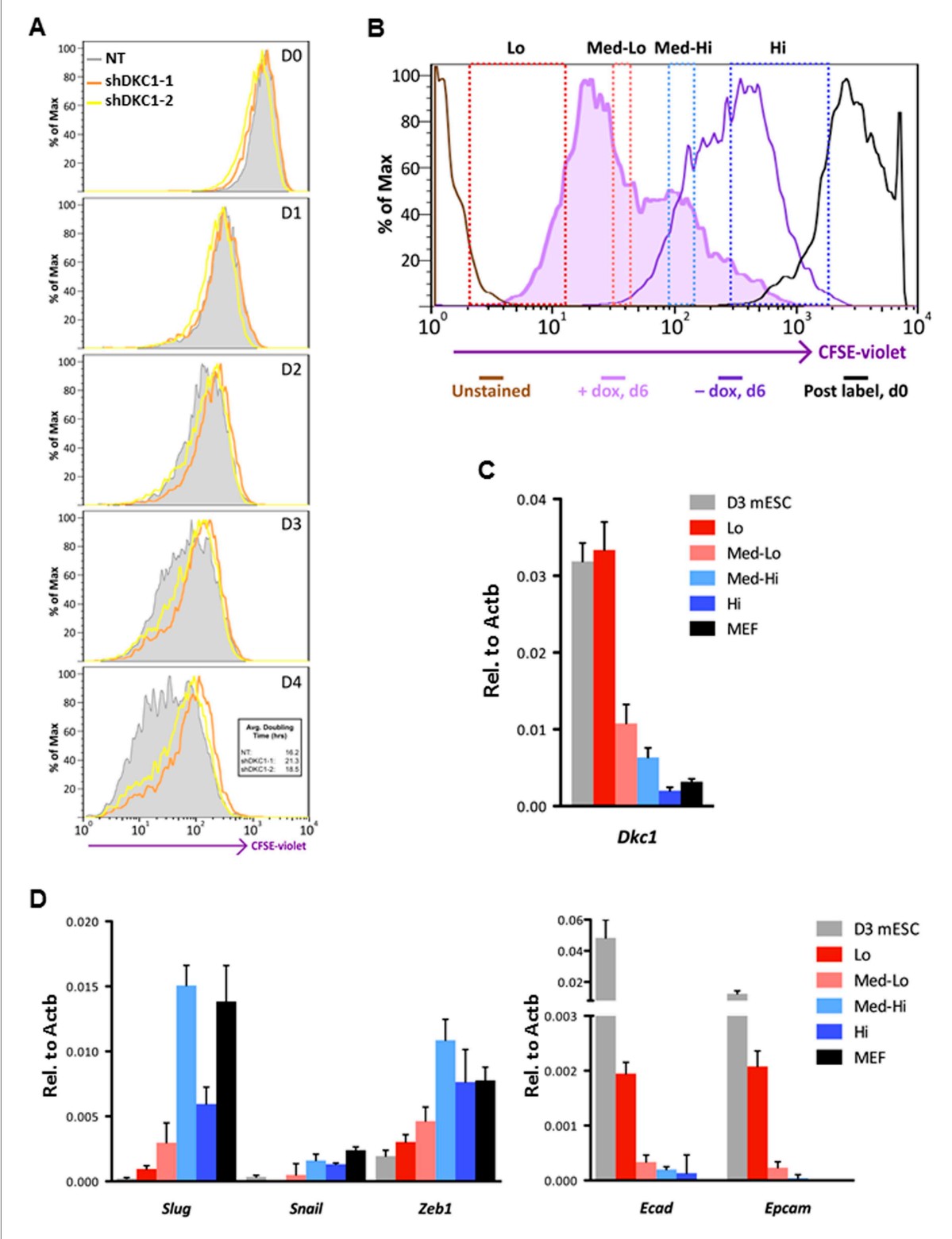

**Figure 8**. Fast cycling somatic cell state conducive to iPS cell generation requires DKC1. (**A**) MEFs depleted of DKC1 by two independent shRNAs (shDKC1-1 and shDKC1-2), along with MEFs infected with control non-target lentiviruses, were analyzed using the CellTrace CFSE Proliferation Assay (Life Technologies). The doubling time for each population was calculated using the mean fluorescence intensity of each timepoint over 96 hr. (**B**) Induced MEFs (light purple) are treated with dox for 4 days, labeled with CFSE, and continuously cultured in the presence of dox for an additional

*Figure 8. Continued on next page*

*Figure 8. Continued*

48 hr prior to FACS. Populations for ultrafast (Lo), fast (Med-Lo), medium slow (Med-Hi), and slow (Hi) cycling dox-induced MEFs are sorted based on CFSE intensity and denoted by dashed boxes. CFSE-intensity of MEFs immediately after labeling (black), unlabeled MEFs (brown), and uninduced MEFs 48 hr post-labeling (dark purple) are shown and used as controls. (**C**) mRNA levels of *Dkc1* and *Oct4* in sorted MEF populations (Lo-Hi) are compared to D3 ES cells and uninduced MEFs by qPCR. Results are normalized to *Actb*. Error bars represent standard deviation (n = 3). (**D**) Ultrafast (CFSE-Lo) cycling MEFs undergo early MET. mRNA levels of mesenchymal genes (*Slug*, *Snail*, and *Zeb1*; left), and epithelial genes (*Ecad* and *Epcam*; right) in sorted CFSE-labeled cell populations are compared to D3 ES cells and uninduced MEFs by qPCR. Data are analyzed as in (**C**). Primer sequences can be found in **Supplementary file 1**.

The following figure supplements are available for figure 8:

**Figure supplement 1**. Ultrafast cycling MEF population contains the bulk of reprogramming activity.

**Figure supplement 2**. Ultrafast cycling somatic cell state can be induced in secondary OKSM MEFs.

**Figure supplement 3**. Ultrafast cycling OKSM MEFs have elevated *Dkc1* levels.

(*Choi et al., 2008*), a finding that remains somewhat controversial in the telomerase field (*Listerman et al., 2014*). In light of our findings that the core DKC1 complex possesses transcriptional coactivator activity, it is tempting to speculate that in the context of WNT-responsive genes, TERT may function to tether the DKC1 complex (as part of the telomerase RNP) to gene promoters and activate transcription by binding to a β-catenin-TCF3 activating complex (*Park et al., 2009*), an integral component of the core stem cell-specific regulatory circuitry (*Cole et al., 2008*). It is, however, unlikely that the coactivator activity detected in our assay is dependent on TERT because we did not detect TERT or any other known components of the telomerase complex (*Fu and Collins, 2007*) in our purified fractions by mass spectrometry (data not shown). This is further supported by our observation that recombinant DKC1-NOP10 heterodimers purified from bacteria was active in transcription. Instead, we favor the model whereby the DKC1 complex (free of any accessory factors) can be recruited to key pluripotency genes via a direct interaction with the XPC complex as we observed in co-immunoprecipitation experiments. This DKC1-XPC assembly is, in turn, recruited to target gene promoters via activator–coactivator interactions with OCT4 and SOX2 (*Fong et al., 2011*; *Gao et al., 2012*). The mechanism by which different activators recruit the same coactivator by targeting distinct subunits or protein surfaces within the DKC1 complex may represent a common strategy that is frequently observed with other transcriptional coactivators such as Mediator (*Taatjes et al., 2002*) and TAFs/TFIID (*Liu et al., 2009*). Therefore, the DKC1 complex may coordinate diverse transcriptional outputs contributing to stemness by cooperating with both stem cell-specific and cell-ubiquitous activators and coactivators (Mediator, the XPC complex, SCC-B). Interestingly, the *Dkc1* gene itself is also a target of OCT4 and NANOG (*Agarwal et al., 2010*). Integrating *Dkc1* into the core regulatory circuitry could further stabilize the autoregulatory loops established by OCT4, SOX2 and NANOG that are postulated to confer stability to self-renewing ES cells without sacrificing their responsiveness to developmental cues during differentiation (*Boyer et al., 2005*; *Loh et al., 2006*).

An increasing number of non-coding RNAs (ncRNAs) have emerged as critical players in gene regulation in both mammals (7SK, Alu, B2) and bacteria (6S) (*Storz et al., 2011*; *Kugel and Goodrich, 2012*). With few exceptions, these ncRNAs function to inhibit the transcriptional activity of their target proteins or protein complexes by forming a stable RNP. The precise mechanism by which snoRNAs may positively or negatively regulate the transcriptional activity of the DKC1 complex is unclear. Additional experiments will be required to determine how the lengths, sequences, and/or secondary/tertiary structures of a subset of snoRNAs may confer coactivator competence to the DKC1 complex. Because as much as 90% of snoRNAs are embedded within introns of protein-coding genes in humans (*Dieci et al., 2009*), their repertoire and relative abundance are directly coupled to the expression of their host genes. Indeed, many snoRNAs are differentially expressed during neural differentiation of mouse ES cells in vitro (*Skreka et al., 2012*). Therefore, it is conceivable that the coactivator activity (and potentially specificity) of the DKC1 snoRNP can be coordinately regulated by cell type-dependent modulation of snoRNA composition.

Considering that many of the disease-causing mutations found in *Dkc1* have been shown to disrupt the binding and/or stability of a select subset of mammalian snoRNAs which we have shown could play

a critical role in conferring coactivator competence to the DKC1 complex, it was somewhat surprising that neither the enzymatic activity nor amino acids mutated in dyskeratosis congenita (DC) patients negatively impacted coactivator activity. Although targeted disruption of *Dkc1* in mice is lethal (*He et al., 2002*), many DC patients carrying mutations in the DKC1 complex live well into their teens and beyond, indicating that these mutations are hypomorphic and can be tolerated during embryogenesis. These findings suggest that amino acid residues in DKC1 critical for transcription are likely largely distinct from those mutated in DC patients. In cases where such mutations do overlap, their effect, if any, on transcription are expected to be subtle because mutations that significantly compromise coactivator function would likely be severely detrimental to the tightly regulated process of mammalian development. However, we note that the various mutant complexes tested in our in vitro transcription assays were reconstituted in insect cells and co-purified with significant amounts of insect snoRNAs (*Figure 4—figure supplement 2*). Because very little is known about the snoRNA repertoire in Sf9 cells and the extent of functional conservation between insect and human snoRNAs, it is unclear how the incorporation of these insect RNAs into human DKC1 complexes might impact structure and function of such hybrid DKC1 RNPs. Therefore, it will be prudent in the future to re-examine these disease relevant mutations in the context of human-protein, human-snoRNA DKC1 RNPs.

Acute depletion of DKC1 in mouse ES cells rapidly down-regulated key pluripotency genes well in advance of telomere attrition and the ensuing cellular senescence that one would expect to occur (after >300 population doublings) due to compromised telomerase function (*Niida et al., 1998*; *Mochizuki et al., 2004*). Taken together with our ChIP results showing specific recruitment of the DKC1 complex to OCT4/SOX2 enhancers of core pluripotency genes in both mouse and human ES cells, as well as the strong dependence of OCT4/SOX2-activated transcription on the DKC1 complex in vitro, we suggest that defects in pluripotency gene expression and stem cell self-renewal upon DKC1 knockdown are at least in part due to compromised transcriptional activation rather than a sole consequence of telomerase deficiency. Given the importance of establishing a robust OCT4/SOX2-depedent transcriptional circuitry during iPS cell induction, it is perhaps not surprising that DKC1 knockdown also severely limits reprogramming capacity of MEFs. Our data, however, indicate that reprogramming of DKC1-deficient MEFs by OKSM aborted at a rather early stage—during the mesenchymal-to-epithelial transition (MET). Specifically, DKC1 appears to be required for the proper induction of epithelial markers like *Ecad* (also known as *Cdh1*) critical for iPS cell generation (*Chen et al., 2010*) even though key negative regulators of epithelial gene expression, *Snail* and *Slug*, were already repressed in DKC1 knockdown MEFs (*Thiery et al., 2009*). Because SOX2 (or KLF4) alone is sufficient to induce *Ecad* expression in MEFs (*Liu et al., 2013*), and OKSM cotarget many MET genes early in the reprogramming process (*Soufi et al., 2012*), the DKC1 complex could cooperate with SOX2 and other reprogramming factors to activate an epithelial gene expression program. It has also been reported that restoring telomerase activity in DC patient-specific fibroblasts carrying a loss-of-function mutation in *Dkc1* (L37del) by overexpressing TERT (*Wong and Collins, 2006*) is insufficient to overcome the reprogramming defect associated with these cells (*Agarwal et al., 2010*). Given that L37del fibroblasts show normal rRNA pseudouridine content as well as rRNA processing kinetics (*Wong and Collins, 2006*), these data taken together strongly suggests a telomerase and ribosome independent mechanism by which the DKC1 complex participates in somatic cell reprogramming. We propose that the DKC1 complex may function to promote the requisite MET during iPS cell generation by activating pro-epithelial genes consistent with its transcriptional coactivator function.

Direct reprogramming of fibroblasts into iPS cells is a slow and presumably stochastic process (*Yamanaka, 2009*). However, accumulating evidence suggests that it is nonetheless amenable to acceleration (and thereby enhanced efficiency) by manipulating pathways that promote cell division (*Banito et al., 2009*; *Hanna et al., 2009*; *Hong et al., 2009*; *Utikal et al., 2009*). These highly proliferative cells competent for reprogramming are also found to exist naturally or become primed by OKSM in a small subset of so called 'privileged' somatic cells within a largely homogeneous population that can proceed through reprogramming in a non-stochastic manner with shorter latency (*Guo et al., 2014*). However, the cell-intrinsic determinants conducive to this privileged state remain unclear, but are likely distinct from factors previously implicated in deterministic cellular reprogramming including SOX2 (*Buganim et al., 2012*) and MBD3 (*Liu et al., 2013*; *Rais et al., 2013*). Here we show that a subpopulation of MEFs proliferating at a significantly faster rate is especially sensitive to DKC1 depletion (*Figure 8A*). Similarly, a small fraction of ultrafast cycling MEFs enriched for *Dkc1* and depleted of mesenchymal signatures also

emerged upon OKSM expression. Although the relationship between these two highly proliferative MEF populations is unclear, we surmise that they could represent a similar privileged somatic cell state. Therefore, the intrinsic variable levels of DKC1 in regular MEFs, or the ability of some MEFs to upregulate *Dkc1* to a critically high threshold level in response to ectopic expression of OKSM, could be a limiting factor in the acquisition of this rare somatic cell state possibly by facilitating an early MET. The precise role of DKC1 in establishing this privileged state in MEFs is unclear but may involve regulating both cellular proliferation and gene expression critical for the early phase of iPS cell formation.

In summary, using an unbiased biochemical approach to probe the transcriptional regulation of the *Nanog* gene by OCT4 and SOX2, we uncovered an unanticipated transcriptional coactivator role of the DKC1 complex and a subset of its associated snoRNAs in ES and iPS cells. We surmise that the DKC1 complex could be one of the cell-intrinsic determinants that impinges on somatic cells during reprogramming by coupling cellular proliferation to stem cell-specific transcription.

## Materials and methods

### DNA constructs and antibodies

cDNAs for human and mouse DKC1, GAR1, NHP2, and NOP10 were obtained from cDNA libraries generated from total RNAs isolated from human NTERA-2 (NT2) and mouse ES D3 cells. Mammalian expression plasmids encoding all four subunits of the DKC1 complex were derived from the pHAGE-EF1α-STEMCCA construct (*Sommer et al., 2009*), wherein OCT4, KLF4, SOX2, and c-MYC were replaced with N-terminal FLAG-tagged DKC1, NHP2, GAR1, and NOP10, respectively (pHAGE-EF1α-DKC1). Expression plasmid for overexpressing the XPC complex (pHAGE-EF1α-XPC) was described (*Fong et al., 2011*). For expressing the DKC1 complex in insect Sf9 cells, N-terminal $His_6$- tagged human DKC1 (wild-type and various disease-associated mutants), untagged GAR1, N-terminal FLAG-tagged NHP2, and untagged NOP10 were inserted into a modified pFastBAC Dual vector (Invitrogen, Carlsbad, CA). For expressing partial and holo DKC1 complexes in *E. coli*, untagged human DKC1, N-terminal HA-tagged GAR1, N-terminal FLAG-tagged NHP2, and C-terminal $His_6$-tagged NOP10 were cloned into a pST44 polycistronic expression plasmid (*Tan et al., 2005*). Of note, GAR1 cDNA was reengineered using Quikchange II Site Directed Mutagenesis Kit (Agilent, Santa Clara, CA) to replace codon-pairs for diglycine residues with sequences that are more favorable for translation (*Li et al., 2012*). For the NAF1-containing intermediate complex, N-terminal HA-tagged NAF1 was inserted in place of HA-GAR1. Polyclonal antibodies against GAR1 (11,711), NHP2 (15,128), FBL (16,021), NOP58 (14,409), and CETN2 (15,877) were purchased from ProteinTech Group; XPC (122A), RAD23B (306A) from Bethyl Laboratories, Montgomery, TX; DKC1 (H-300), TFIIB (C-18), and OCT4 (N-19) from Santa Cruz Biotechnology (Dallas, TX); SOX2 (AB5603) from Millipore (Billerica, MA). Purified rabbit IgGs were purchased from Jackson ImmunoResearch Laboratories (West Grove, PA). Monoclonal antibodies against β-actin (AC-74) were purchased from Sigma Aldrich (St. Louis, MO), DKC1 (H-3) from Santa Cruz Biotechnology, and NOP10 (6547-1) from Epitomics (Burlingame, CA). Anti-FLAG (M2) monoclonal antibodies were purchased from Sigma Aldrich and anti-HA antibodies (MMS-101P) from Covance (Dedham, CA). Antibody against mouse RAD23 was generated in guinea pigs (*Fong et al., 2011*).

### Cell culture

The human embryonal carcinoma NTERA-2 (NT2) cell line was obtained from ATCC. NT2, 293T, and HeLa cells were cultured in DMEM high glucose with GlutaMAX (Invitrogen) supplemented with 10% fetal bovine serum (FBS; HyClone, Piscataway, NJ). Large scale culture of NT2 cells were described (*Fong et al., 2011*). Mouse ES cell line D3 was purchased from ATCC (Manassas, VA) and adapted to feeder-free condition as described (*Fong et al., 2011*). Differentiation of D3 cells was induced by maintaining cells in LIF-free ES cell medium containing 2–5 mM all-trans retinoic acid (Sigma Aldrich) for up to 7 days. Human ES cell line H9 (WiCell, Madison, WI) was maintained in feeder-independent conditions, using Synthemax SC-II Substrate (Corning) and grown in TeSR-E8 (Stemcell Technologies, Canada). Media was changed daily and cell cultures were passaged using Dispase (Stemcell Technologies), according to the manufacturer's protocol.

### Purification of SCC-A/dyskerin Complex

All steps were performed at 4°C. Nuclear extracts were prepared from 400 l of NT2 cells. Partially purified P11-phosphocellulose 1 M KCl and Ni-NTA flowthrough (Ni-FT) fractions were prepared as described (*Fong et al., 2011*). The Ni-FT fraction was dialyzed against buffer D at 0.2 M KCl with

0.0025% NP-40 and 10% glycerol (all buffers from then on contained 0.0025% NP-40 and 10% glycerol unless otherwise stated). This Ni-FT fraction was applied to a Poros 20 HQ column (Applied Biosystems, Carlsbad, CA), subjected to a 4 column volume (CV) linear gradient from 0.2 M to 0.4 M KCl (Q0.3), washed at 0.52 M KCl, and developed with a 13 CV linear gradient from 0.52 M to 1.0 M KCl. Transcriptionally active Q0.3 fraction (0.32–0.4 M) were pooled and applied directly to hydroxyapatite (HAP) type II ceramic resin (Bio-Rad, Hercules, CA), washed first at 0.38 M, then lowered to 0.1 M KCl in 3 CV. HAP column buffer was then exchanged and washed extensively with buffer D at 0.03 M KPi, pH 6.8 without KCl and NP-40. The HAP column was subjected to a 20 CV linear gradient from 0.03 M to 0.6 M KPi. Active HAP fractions eluting from 0.2–0.3 M KPi were pooled and separated on a Superose 6 XK 16/70 gel filtration column (130 ml, GE Healthcare, Piscataway, NJ) equilibrated with buffer D + 0.1 mM EDTA at 0.15 M KCl. Active Superose 6 fractions with an apparent molecular mass of 400–600 kDa were pooled and supplemented with 0.25 mg/ml insulin (Roche, Indianapolis, IN). Pooled fractions were applied to a Poros 20 HE column (Applied Biosystems) equilibrated in buffer D + 0.1 mM EDTA at 0.15 M KCl, subjected to a 34 CV linear gradient from 0.15 M to 1 M KCl. SCC-A containing HE fractions eluted from 0.56–0.62 M KCl. For affinity purification of endogenous DKC1 complexes, Ni-FT derived from 200 l of NT2 cells was applied to a Poros 20 HQ column, subjected to a 22 CV linear gradient from 0.2 M to 1 M KCl. Fractions with low levels of DKC1 were first concentrated using a Spin-X UF concentrator (Corning, Tewksbury, MA) before they were used for immune-affinity purification. Various Poros 20 HQ fractions (adjusted to 0.05% NP-40) were incubated with 10 µg of anti-DKC1 monoclonal antibody immobilized on Protein G Sepharose (GE Healthcare) for 16 hr in the presence of RNase inhibitors (RNasin Plus, Promega, Madison, WI), washed extensively with 0.6 M KCl HEMG buffer (25 mM HEPES, pH 7.9, 0.1 mM EDTA, 12.5 mM MgCl$_2$, 10% glycerol) with 0.2% NP-40, then equilibrated with 0.3 M KCl HEMG with 0.1% NP-40 before elution with peptides.

## Mass spectrometry analysis

Peak Poros 20 Heparin fractions were pooled, concentrated using a Spin-X centrifugal concentrator, separated by SDS-PAGE, stained, protein bands excised, digested with trypsin, and extracted. Peptide pools from each gel slice were analyzed by matrix-assisted laser desorption time-of-flight mass spectrometry (MALDI-TOF MS; Bruker Reflex III). Selected mass values were used to search protein databases linked to PROWL (Rockefeller University) using ProFound and protein databases linked to ExPASy (Swiss Institute of Bioinformatics, Geneva) using PeptIdent.

## In vitro transcription assay

In vitro transcription reactions, DNA template, purification of activators OCT4 and SOX2, general transcription factors, RNA polymerase, and recombinant XPC complex were described (*Fong et al., 2011*).

## 5′ end radiolabeling of RNA

DKC1-associated small RNAs were isolated using TRIzol reagent (Life Technologies, Carlsbad, CA). RNAs were treated with tobacco acid pyrophosphatase (TAP) (Epicentre, Madison, WI) to remove the 5′ m7G cap followed by dephosphorylation with APex Alkaline Phosphatase (Epicentre). Purified RNAs were labeled with T4 polynucleotide kinase (PNK) (New England Biolabs, Ipswich, MA) and γ-$^{32}$P-ATP in the presence of RNase inhibitors (RNAsin Plus, Promega) at 37°C for 1.5 hr. RNAs were precipitated and washed with 75% ethanol to remove free γ-$^{32}$P-ATP. Labeled RNAs were separated on a 6% denaturing Urea-polyacrylamide gel, and visualized by radiography.

## Reconstitution and purification of the DKC1 complexes

Recombinant Bacmid DNAs for expressing wild-type and mutant DKC1 complexes were generated from pFastBAC constructs (described above) according to manufacturer's instructions (Invitrogen). Recombinant baculovirus for the infection of Sf9 cells was generated using the Bac-to-Bac Baculovirus Expression System (Invitrogen). Baculoviruses were amplified three times in Sf9 cells. 2 l of Sf9 cells (~2 × 10$^6$/ml) were infected with baculoviruses, collected at 48 hr post infection, washed once with ice-cold PBS, lysed in six packed cell volume of 0.3 M NaCl buffer HGN (50 mM HEPES, pH 7.9, 10% glycerol, 0.5% NP-40), and sonicated briefly. Cleared lysate was supplemented with 10 mM imidazole and incubated with Ni-NTA resin pre-equilibrated with 0.5 NaCl HGN and 10 mM imidazole for 16 hr. Resin slurries were poured into gravity columns, washed with 0.5 NaCl HGN (0.1% NP-40) with 20 mM imidazole, and bound DKC1 complexes were eluted with buffer 0.3 M NaCl HGN (0.1% NP-40) containing 0.25 M Imidazole. Peak fractions were loaded immediately to a gravity column

containing Heparin Sepharose 6 Fast Flow (GE Healthcare) pre-equilibrated with 0.3 M NaCl HEGN (25 mM HEPES, pH 7.9, 0.1 mM EDTA, 10% glycerol, 0.02% NP-40). Column was washed extensively at 0.3 M NaCl HEGN, then with 0.5 M NaCl HEGN. The DKC1 complexes were eluted with 1 M NaCl HEGN. Peak fractions containing all four subunits of the DKC1 complex, as determined by western blotting, were pooled and incubated with anti-FLAG (M2) agarose (Sigma Aldrich) for 3–4 hr, washed at 0.5 M NaCl HEGN and re-equilibrated with micrococcal nuclease (MNase) digestion buffer (25 mM Tris–HCl, pH 7.9, 20 mM NaCl, 60 mM KCl, 2 mM $CaCl_2$, 0.01% NP-40, 10% glycerol). Bound DKC1 complexes were treated with 300 U of MNase (Thermo Scientific, Waltham, MA) or buffer at room temperature and nutated for 1 hr. MNase digestion was terminated with 20 mM EGTA. Mock and MNase-treated DKC1 complexes were washed extensively with 0.6 M NaCl HEMG with 0.2% NP-40 and 20 mM EGTA and equilibrated with 0.3 M NaCl HEMG with 0.1% NP-40 followed by FLAG peptide elution. For purification of bacterial DKC1 complexes, pST44 expression plasmids were transformed into BL21-Codon Plus RIPL competent cells (Agilent). Expression of hetero-dimeric (FLAG-DKC1/ NOP10-His$_6$), -trimeric (untagged DKC1/FLAG-NHP2/ NOP10-His$_6$), holo (untagged DKC1/HA-GAR1/ FLAG-NHP2/ NOP10-His6) DKC1 complexes as well as NAF1-containing intermediate DKC1 complex (untagged DKC1/HA-NAF1/FLAG-NHP2/ NOP10-His$_6$) were induced at 30°C for 4 hr with 0.5 mM IPTG. Cell pellets were lysed in high salt lysis buffer HSLB (50 mM Tris–HCl pH 7.9, 0.5 M NaCl, 0.6% TritonX-100, 0.05% NP-40, 10% glycerol) with imidazole (10 mM) and lysozyme (0.5 mg/ml). Sonicated lysates were cleared by ultracentrifugation and incubated with Ni-NTA resin for 16 hr. Bound proteins were washed extensively with HSLB with 20 mM imidazole, equilibrated with 0.25 M NaCl HGN (25 mM HEPES, pH 7.9, 10% glycerol, 0.01% NP-40) with 20 mM imidazole, and eluted with 0.25 M imidazole in 0.25 M NaCl HGN. Peak fractions were pooled and applied to a Poros 50 Heparin (HE) column, washed extensively with 0.25 M and 0.5 M NaCl HGN, and subjected to a 4 CV linear gradient from 0.5 M to 1 M NaCl. Fractions containing the desired subunits of the DKC1 complexes were detected by western blotting, pooled, and incubated with anti-FLAG agarose for 3–4 hr at 4°C. Bound proteins were washed extensively at 0.7 M NaCl HGN with 0.1% NP-40 and re-equilibrated with 0.3 M NaCl HGN with 0.1% NP-40 before elution with FLAG peptides. For holo and NAF1-containing DKC1 complexes, HE fractions were first incubated with anti-HA resin, washed and eluted with HA peptides before proceeding to the anti-FLAG affinity immunoprecipitation step as described. 32, 25, 5, and 2 l of *E. coli* cultures were required to generate ~0.5 µg of purified holo DKC1, NAF1 intermediate, hetero-trimeric, and–dimeric complexes, respectively.

## Coimmunoprecipitation assay

pHAGE-EF1α-STEMCCA, pHAGE-EF1α-mXPC, and pHAGE-EF1α-mDKC1 expression plasmids were co-transfected into 293T cells using Lipofectamine 2000 (Invitrogen). Transfected cells on 10 cm dishes were lysed directly on plates with 1 ml of lysis buffer (200 mM NaCl, 50 mM HEPES-KOH, pH 7.9, 0.1 mM EDTA, 0.5% NP-40 and 10% glycerol) 40 hr post-transfection. Cell lysates were collected and homogenized by passing through a 25-gauge needle five times. Lysates were cleared by centrifugation at 15k rpm for 25 min at 4°C. 3 µg of anti-RAD23B antibodies were coupled to Protein A sepharose (GE Healthcare) in PBS containing 0.05% NP-40 for 1 hr at room temperature. Antibody-coupled beads were washed and equilibrated with lysis buffer before incubating with 0.5 ml of cleared cell lysates for 16 hr at 4°C. Sepharose beads were then washed extensively with lysis buffer and bound proteins were eluted with SDS/sample buffer and analyzed by western blotting.

## shRNA-mediated knockdown of DKC1 by lentiviral infection

For lentivirus production, non-target control and pLKO plasmids targeting mouse DKC1 (and XPC) (Sigma Aldrich) were co-transfected with packaging vectors into 293T cells using lipofectamine 2000 (Invitrogen). Supernatants were collected at 48 hr, and again at 72 hr. Virus preparation, titer determination, and infection of D3 mouse ES cells were performed as described (*Fong et al., 2011*), except at a multiplicity of infection (MOI) of 25. For DKC1 knockdown reprogramming experiments, MEFs were transduced at a MOI of 5 prior to iPS cell induction. Detection of alkaline phosphatase activity of knockdown ES cells was carried out using a commercial kit (Millipore).

## Chromatin immunoprecipitation

Mouse ES cell line D3 and human ES cell line H9 were first crosslinked with ethylene glycol bis[succinimidylsuccinate] (EGS, 3 mM, Pierce) for 30 min and then with formaldehyde (1%) for 5 min in fixing buffer (50 mM HEPES, pH 7.5, 0.1 M NaCl, 1 mM EDTA, 0.5 mM EGTA) to capture protein–protein

and protein-DNA interactions (*Zeng et al., 2006*). Crosslinking was then terminated by glycine (0.125 M). Cells were washed twice with PBS, scraped, and centrifuged at 150×g for 5 min at 4°C, resuspended in lysis buffer (50 mM HEPES, pH 7.9, 0.14 M NaCl, 1 mM EDTA, 10% glycerol, 0.5% NP-40, 0.25% Triton X-100) with Halt Protease Inhibitor Cocktail (Pierce, Waltham, MA), and nutated at 4°C for 10 min. Nuclei were pelleted at 1700×g for 5 min, washed twice with wash buffer (10 mM Tris–HCl, pH 8.1, 0.2 M NaCl, 1 mM EDTA, 0.5 mM EGTA) and twice with shearing buffer (0.1% SDS, 1 mM EDTA, 10 mM Tris–HCl, pH 8.1). Nuclei were resuspended in shearing buffer, transferred to Covaris TC 12 × 12 mm tubes with AFA Fiber, and sonicated with a Covaris S2 Focused Ultrasonicator to obtain DNA fragments averaging 300–500 bp in length. Cleared chromatin extracts were adjusted to 0.15 M NaCl and 1% Triton X-100 and immunoprecipitated overnight at 4°C with 3 µg of purified rabbit IgGs or anti-DKC1 antibody. Immunoprecipitated DNA was captured with pre-equilibrated Protein A sepharose (GE Healthcare), washed extensively with high salt wash buffer (0.1% SDS, 1% Triton X-100, 2 mM EDTA, 20 mM HEPES, pH 7.9, 0.5 M NaCl), LiCl wash buffer (100 mM Tris–HCl, pH 7.5, 0.5 M LiCl, 1% NP-40, 1% sodium deoxycholate), and TE buffer (10 mM Tris–HCl, pH 8.0, 0.1 mM EDTA). Supernatant from control IgG immunoprecipitates was saved as input. Input chromatin and immunoprecipitated DNA were reversed crosslinked overnight at 50°C with Proteinase K (Invitrogen), RNase A (Thermo Scientific), and 0.3 M NaCl. DNA was purified using a Qiaquick PCR Purification Kit (Qiagen, Netherlands). Purified DNA was quantified by real time PCR with SYBR Select Master Mix for CFX (Life Technologies) and gene specific primers (*Supplementary file 1*) using a CFX Touch Real-Time PCR Detection System (Bio-Rad). The position of each amplicon relative to transcription start site of mouse *Nanog*, *Oct4*, *Sox2*, *Fgf4*, and human *Oct4* and *Nanog* is indicated in *Figure 5*.

## RNA isolation, reverse transcription and real time PCR analysis

Cells were rinsed once with PBS. Total RNA was extracted and purified using TRIzol reagent (Life Technologies) followed by DNase I treatment (Invitrogen). cDNA synthesis was performed with 1 µg of total RNA using iScript cDNA Synthesis Kit (Bio-Rad) and diluted 10-fold. Real time PCR analysis was carried out with SYBR Select Master Mix for CFX (Life Technologies) and gene specific primers (*Supplementary file 1*) using the CFX96 Touch Real-Time PCR Detection System (Bio-Rad). Results were normalized to β-actin.

## Somatic cell reprogramming and flow cytometry

CF-1 MEFs (Charles River, Wilmington, MA) were transduced with inducible STEMCCA and rtTA lenti-virus-containing supernatants overnight in 8 µg/ml polybrene (Sigma Aldrich). Alternatively, MEFs isolated from mice carrying an integrated dox-inducible transgene expressing OCT4, KLF4, SOX2, and c-MYC (Jackson Laboratories, Bar Habor, ME) were also used. Doxycycline (Sigma Aldrich; 2 µg/ml) was supplemented to complete mouse ES cell media to induce expression of OKSM. Reprogramming was assayed by alkaline phosphatase staining (Millipore), NANOG staining (Abcam, United Kingdom, ab80892), or by flow cytometry analysis using anti-CD90.2/Thy1.2 (Biolegends, San Diego, CA) and anti-SSEA1 (Biolegends, San Diego, CA) on a BD LSRFortessa, performed according to the manufacturers' protocols.

## CFSE labeling of mouse embryonic fibroblasts

To determine the doubling time, MEFs were labeled with CFSE-Violet (Life Technologies) at a working concentration of 5.0 µM, according to the manufacturers' protocol. Cells were analyzed for remaining fluorescence on a BD LSRFortessa every day for 4 days. Induced MEFs were labeled with CFSE-Violet (Life Technologies) at a working concentration of 7.5 µM, as described in *Guo et al., 2014*. CFSE-labeled MEFs were sorted into distinct fast to slow dividing populations at the UC Berkeley Li Ka Shing Flow Cytometry Facility. MEFs cultured in the absence of doxycycline were used as controls.

## Acknowledgements

The authors wish to thank A Fischer at the Tissue Culture Facility (University of California at Berkeley), H Kartoosh at the Li Ka Shing Flow Cytometry Facility (University of California at Berkeley), S Zheng, G Dailey, M Haggart and E Bourbon for technical assistance; S Zhou for mass spectrometry analysis; D King for peptide synthesis; G Mostoslavsky for STEMCCA; M Botchan, K Collins, D Rio, and all members of our laboratory for critical reading of the manuscript. This work was funded by the Howard

Hughes Medical Institute and California Institute for Regenerative Medicine (CIRM) (grant RB4-06016). J Ho is a predoctoral CIRM Scholar (training program TG2-01164).

## Additional information

### Competing interests

RT: Robert Tjian is President of the Howard Hughes Medical Institute (2009–present), one of the three founding funders of *eLife*, and a member of *eLife's* Board of Directors. The other authors declare that no competing interests exist.

### Funding

| Funder | Grant reference number | Author |
|---|---|---|
| Howard Hughes Medical Institute | | Yick W Fong, Jaclyn J Ho, Carla Inouye, Robert Tjian |
| California Institute for Regenerative Medicine | RB4-06016 | Jaclyn J Ho, Robert Tjian |

The funders had no role in study design, data collection and interpretation, or the decision to submit the work for publication.

### Author contributions

YWF, JJH, Conception and design, Acquisition of data, Analysis and interpretation of data, Drafting or revising the article; CI, Acquisition of data, Drafting or revising the article; RT, Conception and design, Drafting or revising the article

## Additional files

### Supplementary file

• Supplementary file 1. Primer sequences.

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
