## [Decision Letter]

Thank you for sending your work entitled “The Dyskerin Ribonucleoprotein Complex as an OCT4/SOX2 Coactivator in Embryonic Stem Cells” for consideration at *eLife*. Your article has been favorably evaluated by James Manley (Senior editor) and 2 reviewers, one of whom is a member of our Board of Reviewing Editors.

The Reviewing editor and the other reviewer discussed their comments before we reached this decision, and the Reviewing editor has assembled the following comments to help you prepare a revised submission.

Fong et al. use biochemical fractionation to isolate co-factors of Oct4/Sox2-mediated, in vitro transcriptional activation of a reporter derived from the Nanog gene. Surprisingly, their fractionation scheme yielded components of Dyskerin (DKC1)-containing RNP complexes, which previously have been ascribed functions in ncRNA biogenesis. The authors show that reconstituted DKC1 complexes consisting of recombinant baculovirus-expressed and bacterial components are also active in Nanog reporter activation, but to a lesser extent than native-purified complexes. ChIP-qPCR assays are employed to assess occupancy of DKC1 over endogenous promoters of pluripotency genes, and to sites co-occupied by Oct4 and Sox2. The authors then employ shRNA knockdown of DKC1 to investigate the impact of its depletion on ES cells and MEF reprogramming. Knockdown of DKC1 causes gross morphological changes to ES colonies and its knockdown in MEFs inhibits reprogramming to iPS cells at an early stage of the process involving an MET-like transition. Finally, the authors propose that DKC1 promotes a “privileged”, rapid-cycling state as a mechanism to support ES cell self-renewal and reprogramming.

This is a potentially interesting study. However, some of the main conclusions are compromised by the lack of adequate specificity controls and further insight into the basis of some of the biological effects the authors are observing. The authors are requested to address the following points:

1) It is not clear how specific the transcriptional effects are in Figures 1, 2 and 3. Do the DKC1-fractions/reconstituted complexes stimulate other pol II reporters, including those that are not associated with pluripotency control? Such controls should be shown.

2) To more convincingly address a role for snoRNA/possible other RNAs associated with DKC1-containing complexes, the authors should treat their native complex with micrococcal nuclease.

3) The use of an additional (protein) cross-linker raises concern as to the extent of the “directness” of specificity of physical interactions by which DKC1 impacts transcription in cells. To address this concern, ChIP-Seq experiments should be performed to assess better the extent and specificity of DKC1 occupancy over genes in ES cells.

4) Related to the last comment, does DKC1 immunolocalize to regions of active chromatin in addition to its concentration in nucleoli and Cajal bodies?

5) The effects of DKC1 knockdown on mRNA expression should be globally assessed by RNA-Seq to ascertain whether the effects on pluripotency gene expression simply reflect more general effects on gene expression.

6) Related to the last comment, does DKC1 knockdown reduce cell doubling rates? This experiment could be performed on mES cells under high LIF concentrations to prevent differentiation.

7) The authors argue that DKC1 depletion blocks induced pluripotency by failing to initiate a mesenchymal-to-epithelial transition (MET). This statement is premature and requires additional experiments.

(i) DKC1 knockdown in MEFs may induce senescence or growth impairment that could also explain the reprogramming defect. The authors need to show a growth curve of DKC1-depleted MEFs in the absence and presence of OKSM expression to rule out this possibility.

(ii) The authors claim that lentivirus-induced DKC1 knockdown blocks iPSC formation. However, they also say that loss of DKC1 impairs ESC self renewal. It is thus important to exclude the possibility that the ”reprogramming phenotype” is a secondary phenotype of differentiation of iPSCs. The authors could either use retroviral vectors to knock down DKC1 since retroviruses become silenced as soon as cells acquire pluripotency. Alternatively, they could follow reprogramming cultures (on dox) at different time points (e.g., day 3, 6 and 9 would be informative) for markers that reflect distinct stages of iPSC formation. This analysis would also give a clearer picture of a possible MET phenotype, which takes place within the first few days of OKSM expression.

8) iPSC colonies are only characterized by AP staining (Figure 7), which is not a stringent marker, and no functional assays of pluripotency are shown (teratoma or chimera formation). At a minimum, the authors should stain a representative plate with a Nanog antibody in order to test whether AP+ colonies are likely to be iPSCs.

---

## [Author Response]

*1) It is not clear how specific the transcriptional effects are in*
Figures 1, 2 and 3*. Do the DKC1-fractions/reconstituted complexes stimulate other pol II reporters, including those that are not associated with pluripotency control? Such controls should be shown*.

We apologize for not referring specifically to this important control. We had previously demonstrated that a partially purified fraction containing all three coactivators (NT2 P1M, see Figure 1—figure supplement 1) is required for OCT4/SOX2 activated transcription of *Nanog* but has no effect on basal or Sp1-activated transcription of a control template in vitro (32). We have now included a statement about the specificity of this SCC activity in the Results section.

*2) To more convincingly address a role for snoRNA/possible other RNAs associated with DKC1-containing complexes, the authors should treat their native complex with micrococcal nuclease*.

We provide compelling evidence in Figure 3 that a wide variety of endogenous DKC1 complexes display dramatically different coactivator activities that cannot be explained by different protein compositions (or posttranslational modifications as determined by SDS-PAGE analysis). This leaves us with the logical assumption that the association of distinct RNA species plays some role in modulating the transcriptional co-activator function of DKC complexes. Furthermore, as shown in Figure 4—figure supplement 2, extensive MNase-treatment of recombinant DKC1 RNPs resulted only on average one cleavage event by saturating amounts of MNase. This suggests that the RNA is likely well protected by the core protein complex and, more importantly, remains stably associated with the DKC1 complex even after MNase treatment. This suggests that even with extensive digestion of RNAs with MNase, the structural integrity of the RNPs remains largely intact. Consistent with this notion, as shown in Figure 4—figure supplement 3, we did not observe significant changes in the activity of the mock and MNase-treated R34W/NOP10 (or WT) recombinant DKC1 complexes, suggesting that a complete digestion of the associated RNA may be required to uncover the full effect of RNA on coactivator activity. Therefore, it is seems unlikely that MNase-treatment of the endogenously DKC1 RNPs would be informative. We could not use other RNases such as RNase A or Benzonase because residual nucleases present in the DKC1 complexes will also degrade mRNAs produced by the in vitro transcription reactions. Moreover, we believe that it will be more informative to use the very limited and precious endogenous purified complexes that we have accumulated for future experiments such as attempts at sequencing the associated RNAs from the most active fractions and compare them to those from least active fractions to potentially identify RNAs critical for DKC1 coactivator function. This experiment is, however, beyond the scope of this first paper.

*3) The use of an additional (protein) cross-linker raises concern as to the extent of the “directness” of specificity of physical interactions by which DKC1 impacts transcription in cells. To address this concern, ChIP-Seq experiments should be performed to assess better the extent and specificity of DKC1 occupancy over genes in ES cells*.

In fact we do not believe DKC1 is directly binding to the promoter DNA and indeed, most transcriptional co-activators do not bind directly to promoter or enhancer DNA. Therefore, as is the case with most coactivators, the DKC1 complex is likely recruited to the promoter by interacting with activators and/or the core promoter machinery. Although not particularly relevant here, using a second cross-linker such as EGS has been reported by others to detect transient interactions of transcription factors with promoter DNA (120). As shown in Figure 5—figure supplement 1, a significant enrichment of DKC1 on the Oct4 distal enhancer is only observed when EGS is used. Importantly, the crosslinking is likely specific as we failed to detect significant enrichment of DKC1 at control regions as well as the oct-sox enhancer of Fgf4 gene. We strongly believe that for the conclusions we make in this paper, the ChIP-qPCR of representative genes is sufficient and that performing ChIP-seq is unnecessary and would add little to this paper but at great cost. We should also point out that in any case our preliminary ChIP experiments indicate that we would likely still require the use of EGS to achieve efficient crosslinking of DKC1 to gene promoters in ESCs.

*4) Related to the last comment, does DKC1 immunolocalize to regions of active chromatin in addition to its concentration in nucleoli and Cajal bodies*?

DKC1 is enriched in the nucleolus but is stained diffusely in the nucleoplasm as well (data not shown and (49)). The DKC1 complex has been shown to be co-transcriptionally recruited to gene promoters to process intron-embedded snoRNAs (21). Taken together with our ChIP-qPCR results and in vitro transcription data, there is a strong case for the recruitment of DKC1 complex to gene promoters acting as a coactivator.

*5) The effects of DKC1 knockdown on mRNA expression should be globally assessed by RNA-Seq to ascertain whether the effects on pluripotency gene expression simply reflect more general effects on gene expression*.

We did not observe gross changes in mRNA levels of a number of control genes including beta-actin (data not shown), GAPDH (data not shown), and T (Brachyury, Figure 6) in DKC1 knockdown ESCs. We have also previously demonstrated in vitro that stem cell coactivators do not enhance basal transcription from the *Nanog* promoter or a “generic” Sp1-dependent promoter (32). Because DKC1 knockdown ESCs undergo spontaneous differentiation, RNA-seq analyses of these cells will likely reveal a transcriptome reflective of the differentiated state wherein many genes whose expression will be down-regulated (i.e. pluripotency genes) or upregulated (i.e. developmental genes) but are not directly regulated by DKC1 (and OCT4/SOX2). Therefore, we do not believe performing RNA-Seq experiment will significantly increase our understanding of DKC1’s role as a coactivator, nor would it definitively address whether DKC1 has a more general effect on gene expression.

*6) Related to the last comment, does DKC1 knockdown reduce cell doubling rates? This experiment could be performed on mES cells under high LIF concentrations to prevent differentiation*.

Since DKC1 knockdown causes rapid differentiation of ESCs, it will be difficult to measure doubling rates while these cells are undergoing differentiation and exiting the cell cycle. In all of our experiments using ESCs, we have been using ∼3 times the amount of recommended LIF. Therefore, we do not believe any further increase in LIF could override the differentiation response caused by DKC1 knockdown in ESCs.

*7) The authors argue that DKC1 depletion blocks induced pluripotency by failing to initiate a mesenchymal-to-epithelial transition (MET). This statement is premature and requires additional experiments*.

*(i) DKC1 knockdown in MEFs may induce senescence or growth impairment that could also explain the reprogramming defect. The authors need to show a growth curve of DKC1-depleted MEFs in the absence and presence of OKSM expression to rule out this possibility*.

As shown in new Figure 8, there is a 2–5 hr lengthening of doubling time of DKC1 knockdown MEFs. However, as discussed in the manuscript, shDKC1-2, which has a more potent effect on blocking reprogramming than shDKC1-1, actually has a milder 2 hr delay in cell cycle (compare to a 5 hr delay with shDKC1-1). Therefore, we do not see a strict correlation between changes in proliferation rate due to DKC1 depletion and reprogramming efficiency. Given that OCT4 and SOX2 have been shown to co-regulate genes essential for MET, we surmise that DKC1 could function as a coactivator to facilitate the expression of OCT4/SOX2-dependent epithelial genes. However, we have now added statements in Results and Discussion sections to point out that cellular proliferation controlled by DKC1 (in addition to regulating gene expression) could also be a contributing factor to efficient iPSC generation. We have also revised the Abstract accordingly.

*(ii) The authors claim that lentivirus-induced DKC1 knockdown blocks iPSC formation. However, they also say that loss of DKC1 impairs ESC self renewal. It is thus important to exclude the possibility that the “reprogramming phenotype” is a secondary phenotype of differentiation of iPSCs. The authors could either use retroviral vectors to knock down DKC1 since retroviruses become silenced as soon as cells acquire pluripotency. Alternatively, they could follow reprogramming cultures (on dox) at different time points (e.g., day 3, 6 and 9 would be informative) for markers that reflect distinct stages of iPSC formation. This analysis would also give a clearer picture of a possible MET phenotype, which takes place within the first few days of OKSM expression*.

We have never observed the appearance of iPSC colonies in DKC1 depleted MEFs that later on regressed during the first 20 days post induction. The majority of DKC1 depleted MEFs failed to undergo MET. For a minor population that was observed to form colonies, they failed to progress further to become SSEA1/NANOG+, as shown in new Figure 7.

*8) iPSC colonies are only characterized by AP staining (*Figure 7*), which is not a stringent marker, and no functional assays of pluripotency are shown (teratoma or chimera formation). At a minimum, the authors should stain a representative plate with a Nanog antibody in order to test whether AP+ colonies are likely to be iPSCs*.

We have now included NANOG immunofluorescence data in new Figure 7 showing the control knockdown MEFs can be reprogrammed to NANOG+ iPSC colonies.